

# Towards a standard typology of endogenous landslide seismic sources

Floriane Provost[1], Jean-Philippe Malet[1], Clément Hibert[1], Agnès Helmstetter[2], Mathilde Radiguet[2], David Amitrano[2], Nadège Langet[3], Eric Larose[2], Clàudia Abancó[4], Marcel Hürlimann[5], Thomas Lebourg[6], Clara Levy[7], Gaëlle Le Roy[2,8], Patrice Ulrich[1], Maurin Vidal[7], and Benjamin Vial[2]

[1]Institut de Physique du Globe de Strasbourg, CNRS UMR 7516, EOST/Université de Strasbourg, 5 rue Descartes, F-67084 Strasbourg Cedex, France.
[2]Univ. Grenoble Alpes, Univ. Savoie Mont Blanc, CNRS, IRD, IFSTTAR, ISTerre, 38000 Grenoble, France.
[3]Norsar, Gunnar Randers Vei 15, NO-2007 Kjeller, Norway.
[4]Geological Hazards Prevention Unit, Institut Cartografic i Geologic de Catalunya, Parc de Montjuïc, SP-08038 Barcelona, Spain.
[5]Departament of Civil and Environmental Engineering, UPC-BarcelonaTECH, C. Jordi Girona 1-3, SP-08034 Barcelona, Spain.
[6]Géosciences Azur, CNRS UMR 7329, OCA/Université de Nice, 250 rue Albert Einstein, F-06905 Sophia-Antipolis Cedex, France.
[7]BRGM, Avenue C. Guillemin, F-45100 Orléans, France.
[8]Géolithe, Crolles, France.

**Correspondence:** Floriane Provost (f.provost@unistra.fr)

**Abstract.** In the last decade, numerous studies focused on the analysis of seismic waves generated by Earth surface processes such as landslides. The installation of seismometers on unstable slopes revealed a variety of seismic signals suspected to be generated by slope deformation, weathering of the slope material or fluid circulation. A standard classification for seismic sources generated by unstable slopes needs to be proposed in order to compare the seismic activity of several unstable slopes and identify possible correlation of the seismic activity rate with triggering factors. The objective of this work is to discuss the typology and source mechanisms of seismic events detected at close distances ($<$ 1 km) and generated by the deformation, failure or propagation of landslides. Seismic observations acquired at 14 sites are analyzed. The sites are representative of various landslide types (i.e. slide, fall, topple, and flow) and material (i.e. from unconsolidated soils to consolidated rocks). The seismic networks installed on these sites are roughly similar (i.e. sensor, network geometry) allowing comparison of the recorded seismic signals. Several signal properties (i.e. waveform, spectral content and spectrogram shape) are taken into account to describe the sources. We observe that similar processes generate similar signals at different sites. A typology is proposed and examples of signals recorded at the different sites are presented. The similarity of the sources and their occurrence for several site configurations make it reasonable to infer the dominant source mechanisms. The proposed typology aims to serve as a reference and a framework for further comparisons of the endogenous micro-seismicity recorded on landslides. The signals discussed in the manuscript are distributed as supplementary material.



# 1 Introduction

Seismology can be used to record (remotely and in a non-invasive way) ground deformation processes and to measure stress/strain conditions through the hydro-mechanical interactions occurring in the media. Seismology is widely used to understand the physical processes taking place on tectonic faults or volcanoes, to investigate fluid reservoir production, and

more recently to analyze the dynamics of Earth surface processes such as glaciers, snow avalanches and landslides. In this manuscript, the term landslide describes a wide variety of processes resulting from the downslope movement of slope-forming materials by falling, toppling, sliding or flowing mechanisms (Hungr et al., 2014). Thus, landslides cover a large range of deformation processes, that can be differentiated in terms of sizes and volumes (smaller than 1 m$^3$ up to larger than 10$^7$ m$^3$), in terms of displacement rates (mm.yr$^-$1 to m.s$^-$1), and in terms of mobilized material (hard/soft rocks, debris, poorly consolidated

soils, and artificial fills).

The analysis of the seismic waves generated by landslides allow monitoring spatio-temporal changes of the stress-strain field in the material from the scale of microscopic internal damage (Dixon et al., 2003; Michlmayr et al., 2012; Smith et al., 2017) to the initiation (e.g. pre-failure) of large ruptures (Amitrano et al., 2005; Yamada et al., 2016b; Poli, 2017; Schöpa et al., 2017). Both the failure and the propagation of the mass generate seismic waves. Physical properties (mass, bulk momentum, velocity,

trajectory) of the landslide can be inferred from the analysis of the seismic signals (Kanamori et al., 1984; Brodsky et al., 2003; Lacroix and Helmstetter, 2011; Ekström and Stark, 2013; Hibert et al., 2014; Levy et al., 2015). Thanks to the increasing number of seismic sensors deployed worldwide and to the development of automatic processing chains, the construction of landslide regional catalogs using seismology is now possible (e.g. Switzerland, Hammer et al. (2013); Dammeier et al. (2016); France, Deparis et al. (2008)). Despite the aforementioned progress in the detection and classification of seismic signals, the

forecast of a particular landslide rupture or acceleration is challenging at the slope scale.

On clayey landslides, drop of shear wave velocity has been observed before acceleration episodes. This shear wave variation through time has been documented using noise correlation techniques for laboratory experiments (Mainsant et al., 2012b), and for a few cases in the field at Pont-Bourquin landslide (Switzerland, Mainsant et al. (2012a)), and at Harmaliére landslide (France, Bièvre et al. (2017)). Precursory seismic signals are also expected and documented before large failures. A precursory

increase in microseismic activity (rate of events and average amplitude) has been observed first before the fall of a coastal cliff (Mesnil-Val, France, Amitrano et al. (2005)) , and was interpreted as the propagation of the fracture. More recently, repeating events have been detected before the Rausu landslide (Japan, Yamada et al. (2016b)) and the Nuugaatsiaq landslide (Greenland, Poli (2017)) . These events are likely associated with the repeated failure of asperities surrounded by aseismic slip, driven by the acceleration of the slope displacement during the nucleation phase of the landslide. Schöpa et al. (2017) recorded harmonic

tremor that started 30 mn before the failure of Askja caldera landslide in Iceland, with temporal fluctuations of resonance frequency around 2.5 Hz. This complex tremor signal was interpreted as due to repeating stick-slip events, with very short recurrence times (less than 1 s), producing a continuous signal. In these three studies, there were no nearby sensor (< 1 km), preventing an accurate location and characterization of these signals. Therefore, the monitoring of endogenous micro-seismicity may represent a promising approach especially, with the advent of robust, cheaper and portable seismic sensors and digitizers.

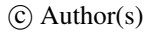



It is now possible to install dense sensor networks close to the unstable slopes and record low amplitude signals in broad frequency bands. A wide variety of unstable slopes are currently monitored (i.e. through permanent or campaign installation) with seismic networks of different sizes and instruments (Table 1). The next sections discuss the seismic instrumentation installed on landslides since the 1960s.

## 1.1 Historical implementation of Acoustic Emission (AE) and Micro-Seismicity (MS) instrumentation

In the 1960s, Cadman and Goodman (1967) observed an increase of Acoustic Emissions (AE) generated by slopes tilted towards failure at both laboratory and field scales. AEs are high frequency (10-1000 kHz) body waves generated by the release of strain energy through grain rearrangement (Michlmayr et al., 2012). Further studies confirmed these results for several slopes (Rouse et al., 1991; Amitrano et al., 2005; Smith et al., 2014; Dixon et al., 2015) where correlations between AE, surface displacement and heavy rainfall were observed . AE can record deep deformation processes before signs of displacement are identifiable at the surface . However, AEs are rapidly attenuated with the distance to the sources. The location of the sensors and the type of waveguide are also critical to capture the slope behavior. Recent developments of Fiber Optic Distributed Acoustic Systems (FO-DAS) offer the opportunity to overcome attenuation limitations and deploy measures over long distances (Michlmayr et al., 2017).

More recently, several studies investigated the micro-seismicity (MS) observed on unstable slopes. MS studies analyze the seismic waves generated in low frequency ranges (1-500 Hz) by the release of strain energy in the ground at larger scale than the grain to grain interactions. The method offers the opportunity to remotely record the spatial distribution of the deformation through time (McCann and Forster, 1990; BRGM, 1995) and is less sensitive to attenuation than AE methods. Gomberg et al. (1995) installed seismometers on the Slumgullion (Colorado, USA) slow-moving landslide, to describe the various recorded signals and to characterize the sources in order to retrieve the mechanical processes taking place during landslide deformation. Further studies used the same method for several slope configurations (hard/soft rocks, soils, very slow to rapid movements) but also investigated the possible links between the displacement rate and the seismic energy release (Spillmann et al., 2007; Helmstetter and Garambois, 2010; Walter et al., 2012, 2013b; Tonnellier et al., 2013). Helmstetter and Garambois (2010) correlated the seismic response of the Séchilienne rockslide with the surface displacement rate and rainfall amount.

## 1.2 MS processing chains

One of the current challenges for landslide MS analysis is the development of dedicated processing chains able to analyze the unconventional seismic signals observed on landslides. The three steps of MS processing are successively: the detection, the classification and the location of the endogenous seismic events. The development of robust and versatile processing chains for analyzing landslide micro-seismicity is challenging because of 1) the small magnitude of the events and the attenuation of the media that results in emergent and low Signal-to-Noise Ratio (SNR) records, 2) the seismic source radiation patterns that may be single centroid source, double couple source or volumetric source, and, 3) the heterogeneity and variation in time (i.e. topography, water table levels, fissures) of the underground structure preventing the construction of precise velocity models and hence, accurate source locations. Regarding the first challenge of detecting the events, the use of spectrograms to detect





manually or automatically seismic events is common. Spectrograms (or sonograms) represents the evolution of the frequency content in time by computing the Fourier Transform on small moving time windows (e.g. < 1 s). Automatic detection is usually achieved with the STA/LTA (Short-Term Average/Long-Term Average) detector (Allen, 1982) applied on the summed energy of the spectrogram (Spillmann et al., 2007; Helmstetter and Garambois, 2010; Tonnellier et al., 2013). Second, classifying

the detected signals can then be carried out automatically by discarding exogenous events with simple criteria (i.e. threshold on the signal duration, inter-trace correlation, apparent velocity). Machine learning algorithms offer nowadays the possibility to automatize and improve this step. (Dammeier et al., 2016) developed a Hidden Markov Model (HMM) that can detect automatically in the time series the occurrence of one particular type of events. The success rate of HMM is reasonable and this technique has the advantage of requiring only one single example to scan the time series . The Random Forest algorithm

has proven its efficiency for volcanic and landslide signals classification with higher success rate and versatility (Provost et al., 2017; Hibert et al., 2017c). New signals are successfully classified in multiple pre-defined classes and changes in the source properties may be detected by change on the uncertainties (Hibert et al., 2017c). It must be noticed that this approach requires a training set with sufficient elements to build the model. Good success rates (i.e. > 85 %) are rapidly reached with 100 elements or more per class.

Finally, the location of the sources is the final and probably most challenging step. Common location methods (such as NonLinLoc) were used in combination to 3D-velocity models for locating impulsive micro-earthquakes occurring at the Randa rockslide (Spillmann et al., 2007). However, a certain number of recorded signals do not exhibit impulsive first arrivals and clear P and S-waves onsets. For this kind of signal, location methods based on the inter-trace correlation of the surface waves (Lacroix and Helmstetter, 2011) or on the amplitude (Burtin et al., 2016; Walter et al., 2017) are more suitable and easier to

automatize. Other methods such as HypoLine (Joswig, 2008) aim at integrating different strategies (i.e. first arrival picking, inter-trace correlation and beam-forming) to locate accurately the epicenter under the control of an operator. Provost et al., 2018 (in press) aimed at combining Amplitude Source Location (ASL) and inter-trace correlation of the first arrivals in an automatic scheme. This strategy showed accurate location of impulsive events while the error on the epicenter of emergent events is reduced by the use of ASL to constrain the location. Many studies approximate the media attenuation field and/or

the ground velocity, or do not take into account the topography, leading to mis-location of the events that prevents for accurate interpretation of certain sources and leads to false alarms (Walter et al., 2017).

Notwithstanding the number of studies on landslide MS, understanding the possible mechanisms generating these signals needs to be achieved. The discrimination of the endogenous landslide seismic signals is difficult and need to be established. The objective of this paper is thus to propose a typology of the landslide micro-seismic signals from seismic records in the field. The

proposed typology is based on the analysis of records from 13 monitored sites. The typology includes all the seismic sources recorded at near distances (< 1 km) and in the frequency range of MS studies (1-500 Hz) generated by landslides 1) developed in hard/soft rocks and soils, 2) characterized by fragile (i.e. rupture) and ductile (i.e. viscous) deformation mechanisms.

In this study, we first discuss all the physical processes that occur on landslides and may generate seismic signals. We further present the seismologically-instrumented landslides in the world and describe the instrumentation. Then we establish

a classification scheme of the landslide seismic signals from relevant signal features. We further discuss the perspectives and



remaining challenges of monitoring landslide deformation with MS approaches. The seismic signals associated with very large rock/debris avalanches observed at regional distances are out of the scope of this work.

## 2 Description of landslide endogenous seismic sources

This section describes the possible mechanical processes observed on landslides and susceptible to generate seismic sources.
We present the conditions surrounding their occurrences (type of material, topography), their sizes, and their mechanical properties.

### 2.1 Fracture related sources

The term fracture denominates any discontinuous surface observed in consolidated media and originating from the formation of the rocks (i.e. joint) or the action of tectonic (i.e. schistosity), gravitational or hydraulic loads. In the case of slow-moving
landslides, the propagation of the material also creates fractures on the edge and at the base of the moving material. Fractures occur in all type of materials at different scales from grain rupture to metric faults. The term fissure is sometimes used to describe fractures affecting the surface of the ground and for fractures affecting poorly consolidated material. We here include all these surface discontinuities under the general term of "fracture".

Fractures are generated in three basic modes (I: opening, II: sliding and III: tearing) depending on the movement of the
medium on the sides of the fracture plane. They result from either brittle failure of the media or from dessication effects forming polygonal failures during soil drying. On landslides, most of the fractures occur in a tensile mode because of the low tensile toughness of the landslide material and the shallow depth (Stumpf et al., 2013). The formation of fractures can also be generated in depth by progressive degradation of the rock through ground shaking and/or through weathering and long-term damage due to gravitational load. At the base and on the edges of the landslide, the movement is assumed to develop fractures
in shear mode, creating sliding surfaces.

Shearing on the fracture plane and tensile fracture opening/closing generate seismic signals. Shearing takes place at different scales from earthquakes on tectonic plates to grain friction and generates a variety of seismic signals (Zigone et al., 2011). Unstable regime leads to stick-slip behavior where the stress is regularly suddenly released generating impulsive seismic events. Tremor like signals or isolated impulsive or emergent events are also generated during plate motions. This variety of signals are
observed during glacier motion. Deep icequakes are usually associated to basal motion (Winberry et al., 2011; Pratt et al., 2014; Helmstetter et al., 2015a, b; Röösli et al., 2016a; Podolskiy and Walter, 2016). Tremor like signals are also recorded during glacier motion (Lipovsky and Dunham, 2016). They are characterized by long duration signals of low amplitudes with no clear phase onsets. They are associated with repetitive stick-slip events on the fracture plane. Tensile fracture opening/closing generate similar signals on glacier at the surface and at depth (Walter et al., 2013a; Helmstetter et al., 2015b; Podolskiy and
Walter, 2016). Focal mechanism and location of the source allow to differentiate between tensile and shear mechanism.



## 2.2 Topple and fall related sources

On vertical to sub-vertical slopes, mass movement occurs as the topple of rock columns or as the free-fall (and possibly bouncing and rolling) of rocky blocks (Hungr et al., 2014). In the case of toppling, the movement starts with a slow rotation of the rock blocks under the effects of water infiltration or ground shaking and ends with the free fall of larger blocks. Rockfalls,

during the propagation phase, impact the ground at some location along their trajectory. These impacts generate seismic waves that can be recorded remotely by seismometers. The range of rockfall volumes can be very large, varying from less than one cubic meter to thousand cubic of meters.

## 2.3 Mass flow related sources

Mass flows gather different run-out processes of debris or of a mixture of water and debris. They cover a large range of

volumes from large rock avalanches of several millions cubic meters to small (hundreds cubic of meters) debris flows (Hungr et al., 2001). They can occur in wet or dry conditions on low to steep slopes. The contacts of the rock/debris fragments with the bedrock and in the mass flow generate seismic radiations (Suriñach et al., 2001; Burtin et al., 2009; Schneider et al., 2010; Hibert et al., 2011; Abancó et al., 2012; Burtin et al., 2013; Levy et al., 2015; Vázquez et al., 2016; Hibert et al., 2017b). The seismic signal is hence a combination of grain contacts within the granular flow and of grain to ground surface contacts and

hence generate a complex seismic signal.

## 2.4 Fluid related sources

Hydrological forcing (e.g. precipitation, snow-melt) is one of the most common landslide trigger. The presence of fracture networks, water pipes and the heterogeneity of the rock/soil media result in the development of preferential water flow paths (Richards and Reddy, 2007; Hencher, 2010). These preferential flows induced local saturated area where the increase of pore

water pressure may destabilize shallow or deep shear surfaces. In soils, the dissolution of material into finer granular debris creates weak zones prone to collapse either by suffusion (i.e. non cohesive material wash out under mechanical action) or by dispersion (i.e. chemical dissolution of fractured clay soils; Richards, Jones, 1981). In rocks, pipes may develop by erosion. In these saturated fracture networks, hydraulic fracturing can occur creating earthquakes and harmonic tremors related to flow migration in the fractures (Chouet, 1988; Benson et al., 2008; Tary et al., 2014a, b; Derode et al., 2015; Helmstetter et al.,

2015b).

## 3 Landslide seismological instrumentation

### 3.1 Sensors used in landslide monitoring

Body and surface mechanical waves may be generated by the sources described in Section 2. Body waves (Primary -P-, Secondary -S-) radiate inside the media. P-waves shake the ground in the same direction they propagate while S- waves

shake the ground perpendicularly to their propagation direction. Surface waves only travel along the surface of the ground and




their velocity, frequency content and intensity change with the depth of propagation. Acoustic waves can be generated by the conversion of body waves at the surface. These waves travel in the air at a velocity of about 340 m.s$^{-1}$, slightly varying with temperature and pressure. Acoustics wave are often generated by anthropic or atmospheric sources (e.g., gun shots, explosions, storms...), but can also be generated by rockfalls, debris flows or shallow fracture events. All these mechanical waves are subject

to attenuation with the travel distance; high frequency being attenuated faster than low frequency waves. The relatively low energy released by the landslide related sources makes the choice of the seismological instruments to deploy very important. Four types of instruments are used to record ground motion for different frequency ranges and sensitivities. For landslide monitoring, Short-Period (SP) seismometers and geophones, Broad-Band (BB) seismometers, accelerometers, and AE sensors are commonly installed in the field.

– Broad-Band seismometers are force-balanced sensors with very low corner frequency (< 0.01 Hz) that can record the ground motion with a flat response in a large frequency range [0.01-25] Hz. They require a careful mass calibration during their installation and are sensitive to temperature and pressure variations. They are mostly used to record very weak ground motion and ambient noise;

    – SP-seismometers are passive or force-balanced instruments with high corner frequency (> 1Hz). They measure the ve-

locity of the ground with high sensitivity and a flat response in the [1-100] Hz frequency band. They are recommended for volcanic and glacier monitoring among other applications. They are less sensitive to temperature and pressure variations and do not require mass calibration. They are hence particularly suitable for landslide monitoring. Geophones are similar to SP seismometers but usually cover higher frequencies [1-600]Hz with lower sensitivity. They are mainly used for active seismic campaigns but may also be installed for the same purposes as SP seismometers;

– Accelerometers are strong motion sensors able to record high amplitudes and high frequencies seismic waves. They can resolve accelerations in the frequency bands from 0.1 to 10 kHz. The response of the sensor is proportional to ground acceleration for all frequencies (there is no corner frequency). But the noise level is important for low frequencies and the sensitivity is not as good as for velocimeters. They are used to record strong ground motion in particular when installed close to epicenters (< 100 km) of large earthquakes where seismometers usually saturate. For landslide, they are usually

used as inclinometers;

    – AE sensors can record ground vibrations at very high frequencies (10 kHz-10 MHz) and low amplitude. There are two types of AE sensors: the first type is very sensitive to a narrow frequency band only while the second is sensitive to a broader frequency band (Michlmayr et al., 2012). In the field, a waveguide is often installed together with AE sensor in order to counteract the attenuation of the signal. They are very often used in combination with accelerometers for

structural monitoring and for laboratory experiments (e.g. loading, shear, flume tests) and can be used on landslide to monitor very low magnitude sources at the grain-to-grain interactions (Dixon et al., 2003; Michlmayr et al., 2012; Smith et al., 2017);



- Microphones can be useful, in addition, to detect, locate and classify landslides seismic signals (Helmstetter and Janex, 2017). The detection of acoustic waves and body waves at one point, because they propagate at different velocities, can be used to estimate the distance from the source. The relative amplitude of seismic and acoustic waves can also provide information on the depth of the source, because shallow sources generate more acoustic waves than deeper ones.

It must be noted that AE sensors only record acoustic emissions generated at less than few meters from the source due to the high attenuation of the high-frequencies. Even with a waveguide, they must be collocated with the cracks or the sliding surfaces observed on the slope. On the contrary, seismometers and geophones offer a solution to monitor more distant sources. BB seismometers can be used to explore the low-frequency signals. They also record higher frequencies although Geophones and SP seismometers are more adapted and cheaper. Dense networks of the latter instruments are recommended to investigate
the seismicity induced by landslide deformation while the installation of one BB seismometer is enough to investigate the low-frequency radiations of the landslide.

## 3.2 Data

Seismic networks have been installed on several unstable slopes worldwide since the initial study of Cadman and Goodman (1967). Table 1 synthesizes the unstable slopes or catchments instrumented with seismological sensors worldwide. The sites are classified in terms of landslide types (i.e. slide, fall and flow) according to (Cruden and Varnes, 1996). Studies on snow
avalanches (Lawrence and Williams, 1976; Kishimura and Izumi, 1997; Sabot et al., 1998; Suriñach et al., 2001; Biescas et al., 2003) are not integrated. Most of the instrumented sites are located in the European Alps (France, Italy and Switzerland). Short-Period (SP) seismometers and Geophones (G) are the most common type of instruments. Their installation and maintenance is easy as they do not require mass calibration in comparison to Broad-band (BB) or long-period (LP) seismometers. Seismolog-
ical networks installed on unstable slopes are often designed as terms of clustered arrays of a minimum of four seismometers. The common geometry of these types of arrays consists in one central three component SP seismometer at the center of three (or more) vertical component SP seismometers. This geometry enables a better identification of the source azimuth and of the apparent velocity with a Beam-Forming location method (Joswig, 2008; Lacroix and Helmstetter, 2011; Walter et al., 2012; Tonnellier et al., 2013; Vouillamoz et al., 2017).
Seismological observations from 14 sites have been used to propose the typology (Table 1). The sites are representative of various types of slope instabilities and rocks. For all sites, the instruments are deployed close to the landslide. The instruments are mainly SP seismometers with short sensor-to-sensor distances and cut-off frequencies of 1 Hz to 5 Hz. Certain networks are deployed permanently and are used as reference to document the evolution of the seismogenic landslide activity over time (e.g. France, RESIF/OMIV (2015) ; Switzerland, Spillmann et al. (2007); Walter et al. (2017)). Other networks are deployed
for campaign measurements in order to document specific activity periods (e.g. acceleration).



## 4 Processing methodology

Seismic observations can be analyzed using two approaches, e.g. micro-seismicity analysis and seismic noise analysis. The first approach consists in cataloging the seismic signals triggered by the slope deformation, locating these sources and correlating the spatio-temporal occurrence with different deformation patterns. The second approach consists in analyzing the seismic noise. The resonance frequency of the noise can be related to the rigidity of the unstable mass and thermal forcing and the correlation between different sensors can be used to estimate the surface wave velocity and its evolution in time (Larose et al., 2015, and references therein).

In order to propose a reference typology of landslide endogenous seismic sources when using the first approach, seismic signals recorded at different sites are compared. The classification of the landslide seismic sources is based on the description of the signals with nine parameters:

- the duration of the signal $T$ (expressed in second), computed on the stacked spectrogram of the traces (Helmstetter and Garambois, 2010).

- the dissymetry coefficient of the signal (expressed in percent), computed as:

$$s = \frac{t_m - t_1}{t_2 - t_1} \times 100 \tag{1}$$

with $t_1$, $t_2$ and $t_m$ the time of the signal onset, ending and maximum respectively.

- the number of peaks of the signal envelop $N_{peaks}$, computed as the number of local maximum above 50% of maximal value of the signal envelop. The envelop of the signal is computed as the absolute value of the Hilbert transform of the signal. The envelop is smoothed by a computing the average of on a moving window of length: $\delta t = \frac{100}{f_s T}$.

- the duration of the signal auto-correlation, defined as:

$$A_{max} = \frac{t_c}{T} \tag{2}$$

with,

$$t_c = \max_t(C(t) < 0.2 * max(C)) \tag{3}$$

with $C$ equal to the signal auto-correlation. $A_{max}$ is expressed in percent (%) and represents the duration of the signal correlating with itself. As an example, a signal with a rapid and abrupt change in frequency content will rapidly be uncorrelated (low $A_{max}$) while a signal with a constant frequency content will have a long auto-correlation (high $A_{max}$).

- the mean frequency (expressed in Hertz), computed as:

$$F_{mean} = \frac{\sum_{i=1}^{N} PSD(f_i) f_i}{\sum_{i=1}^{N} PSD(f_i)} \tag{4}$$



with the Power Spectral Density (PSD) defined as:

$$PSD(f) = \frac{2|FFT(y)|^2}{Nf_s} \tag{5}$$

with $f_s$ and N being the sampling frequency of the signal and the number of samples respectively. The mean frequency is chosen here as it more representative of the signal spectrum energy and less sensitive to noise than the frequency of maximum energy. (Farin et al., 2014).

– the frequency corresponding to the maximal energy of the spectrum $F_{max}$ (expressed in Hertz).

– the frequency bandwidth $F_w$ defined as:

$$F_w = 2\sqrt{\frac{\sum_{i=1}^{N} PSD(f_i)f_i^2}{\sum_{i=1}^{N} PSD(f_i)} - F_{mean}^2} \tag{6}$$

– the minimal frequency of the signal spectrum, computed as:

$$f_{min} = \min_{f}(PSD(f) < 0.2 \times \max(PSD)) \tag{7}$$

– the maximal frequency of the signal spectrum, computed as:

$$f_{max} = \max_{f}(PSD(f) < 0.2 \times \max(PSD)) \tag{8}$$

the maximal frequency of the signal spectrum $f_{max}$ (not to be confused with parameter $F_{max}$ defined above).

The attributes are always computed on the trace with the maximal amplitude band-passed in the range [1-50] Hz enabling both to limit the influence of the wave propagation and to compare signals with different sampling frequencies (i.e 120 Hz to 1000 Hz). These nine parameters are chosen because they correspond to the criteria used by experts to analyze and classify a seismic signal and also because they can be used in automatic classification algorithms (Fäh and Koch, 2002; Langer et al., 2006; Curilem et al., 2009; Hammer et al., 2012, 2013; Hibert et al., 2014a; Ruano et al., 2014; Maggi et al., 2017; Provost et al., 2017; Hibert et al., 2017c). They can be computed for any signal types and present a robust framework for future comparison. Moreover, recently use of Random Forest algorithm makes it possible to confirm the utility of this choice (Provost et al., 2017; Hibert et al., 2017c). Most of these parameters are dependent on the source sizes, the source to sensor distances and the media properties (attenuation, dispersion).

## 5 Seismic description of the signals - typology

The typology of the signals is mainly based on the duration and the frequency content of the seismic signals. The signals are classified in three main classes: "Slopequake" (SQ), "Rockfall" (RF) and "Granular flow" (GF) signals. For "Slopequake", sub-classes are proposed and discussed based on the frequency content of the signals. Several examples of signals recorded at different sites are presented and the sources are discussed in the corresponding section.



## 5.1 Rockfall (RF)

Figure 3 displays the seismic waves recorded for a single rock fall at the Riou-Bourdoux catchment (French Alps; Hibert et al. (2017a)). The signal is characterized by successive impacts visible both on the waveform and on the spectrograms. Depending on the height of the cliff, the signal lasts between 5 s and tens of seconds. The spectral content contains mostly frequencies
above 10 Hz. However, energy below 10 Hz is present for certain impacts for rocks with volumes larger than 1 m$^3$ (Figure 3a). At closer distance, very high frequencies can be recorded up to 100 Hz (Figure 3a). Theoretically, the corner frequency of such events is expected between 100 Hz and 500 Hz depending on the attenuation of the media (Farin et al., 2014) but in most of the cases the attenuation of the medium eliminates frequencies greater than 100 Hz. The auto-correlation remains large over time due to the similitude of the individual impacts signals ($T_{corr} > 10\%$). P- and S- waves are hardly distinguishable on the
record and the signals recorded at the seismic sensors are dominated by surface waves (Deparis et al., 2008; Dammeier et al., 2011; Helmstetter and Garambois, 2010; Hibert et al., 2014; Levy et al., 2015). The first arrivals are mainly impulsive.

    Seismic signals of natural masses detaching from cliffs are presented in Figure 4. They present similar characteristics to the artificially triggered rockfall. The highest measurably frequency depends on the source-to-sensor distance. Generally, the Power Spectral Density energy is low below 10-15 Hz. The initial falling masses can themselves broke into smaller units
during propagation. In this case, the signal does not return to the noise level between the impacts due to developing granular flow (Figure 4.b,e,f). When several blocks are falling at the same time, impacts may overlap, so do the peaks of the signals. In certain cases, the first rock free-fall is preceded by a signal that can be associated with the rock detachment. An example of this precursory signal can be observed in Figure 4.a,f and in (Hibert et al., 2011; Le Roy et al., 2017; Dietze et al., 2017b). The seismic signals of rockfalls contain information on the physics of these phenomenon. The seismic energy of rockfall signals
has been shown to be proportional to the volume (Hibert et al., 2014a; Farin et al., 2014; Le Roy et al., 2018). Scaling laws are also shown in the case of block falls between seismic energy, momentum, block mass and velocity before impacts (Hibert et al., 2017a). The frequency content is also controlled by the block mass (Farin et al., 2014; Burtin et al., 2016). If the block falls are well isolated, each impact generates impulsive waves. In the case of multiple block falls or short distances between the seismic source and the sensor, the first arrivals may be emergent due to the simultaneous arrivals of the waves.

## 5.2 Granular Flow (GF)

Granular flows are characterized by cigare-shape signals lasting between tens to thousands of seconds. They are subdivided in two classes:

- **Dry Granular flow** (Figure 5): They are characterized by cigare-shape waveforms of relatively long duration ($< 500$ s). Due to the absence of water the source generally propagate on small distances. No distinguishable impacts can be
30       observed in the waveform nor in the spectrogram, in contrast to rockfall signals. The dissymetry coefficient of the signal varies between 30% and 75% and depends on the acceleration and the volume of mass involved in the flow through time (Suriñach et al., 2001; Suriñach et al., 2005; Schneider et al., 2010; Levy et al., 2015; Hibert et al., 2017b). The autocorrelation decreases rapidly in the first third of the signal duration. The frequency ranges from 1 to 35 Hz. The



mean frequency of the PSD varies between 5Hz and 10 Hz and can be larger (up to 20 Hz) when the seismic sensors are located close to the propagation path. The auto-correlation is very weak h ($T_{corr} \approx 0\%$) and no seismic phase can be distinguished.

- **Wet Granular flow** (Figure 6): These signals last several thousands of seconds to several hours and correspond to debris flows. They occur during rainfall episodes when fine material and boulders run down the stream over long distances ($> 500$ m). The seismic sensors are often installed at very close distance to the flow path so high frequencies up to 100 Hz may be recorded (Abancó et al., 2014; Burtin et al., 2016; Walter et al., 2017). Little energy is present in the low-frequencies ($< 10$ Hz) depending on the amount of water and the size of the rock blocks involved in the flow (Burtin et al., 2016). The signal is emergent and the amplitude variation corresponds to the mass involved in the flow passing in the vicinity of the sensor. Debris flow are very often divided in a front with the largest boulders and highest velocity followed by a body and a tail where the sediment concentration and the velocity progressively decreases (Pierson, 1995). The seismic signal amplitude hence increases progressively as the front is passing in the vicinity of the sensor (Abancó et al., 2012; Hürlimann et al., 2014; Burtin et al., 2016; Walter et al., 2017) and decreases progressively, as the front is moving away from the sensor. Large spikes and low-frequencies may be observed in the seismic signal corresponding to the front of the debris flow generated by large boulders impacts. The frequency content also changes and progressively, energy in the lower frequencies decreases (Figure 6.a). The auto-correlation is very weak ($T_{corr} = 0\%$) and no seismic phase can be distinguished.

## 5.3 Slopequake (SQ)

The "Slopequake" class gathers all the seismic signals generated by sources located within the slope at its sub-surface or at depth such as fracture related sources or fluid migration (cf. section 2). They are mainly characterized by short duration ($< 10$ s). They are sub-divided in two classes "Simple" and "Complex".

### 5.3.1 Simple Slopequake

The first class "Simple Slopequake" of short duration signals is characterized by short ($< 2$ s) to very short duration ($< 1$ s) signals. Their main characteristic is the triangular-shape of the spectrogram. The first arrivals contain the highest frequencies of the signal and are followed by a decrease of the frequencies. Depending on the frequency content, these signals can be sub-divided into three classes:

- **Low-Frequency Slopequake (LF-SQ)** (Figure 7): The signal last between 1 and 5 s. The maximal amplitude of the signal waveform occurs at the beginning or at the center of the signal ($15\% < s < 50\%$). The waveform presents only one peak and most of the first arrivals are emergent. Phase onsets are difficult to identify. The signals seem to be dominated by surface waves.

- **High-Frequency Slopequake (HF-SQ)** (Figure 8)): The signal is very brief ($< 1$ s) and energetic. The maximal amplitude of the signal waveform occurs close to the beginning of the signal ($s < 30\%$). The waveform presents only one peak



and most of the first arrivals are impulsive. Although, the beginning of some of the signal becomes emergent with the distance and the maximal amplitude is shifted to the center of the signal (Figure 8). Different phases may be observed (Spillmann et al., 2007; Lévy et al., 2010): P-arrivals are detected at the beginning of the signal and correspond to the high frequency waves, surface waves are then observed at the time the frequency decreases. In most of the cases, the picking of the different waves onset is made difficult because of the sensor to source distances. The travel-time difference between the different wave onsets is very short ($< 20$ ms) in most of the cases and body and surface waves may be difficult to identify. It results from the fact that most of the studied landslides (especially for soft-rock landslides) present shallow basal surfaces and most of the sources are very weak ($M_L <0$) so they can only be recorded at really close distance.

– **Hybrid Slopequake (Hybrid-SQ)** (Figure 9)): The signal last between 1 and 2s. It presents the characteristics of the two precedent signals. The brief first arrivals are very impulsive and last less than one second. They are followed by a low-frequency coda similar to the Low-frequency slopequake. The maximal amplitude of the signal waveform occurs close to the beginning of the signal (s $< 40\%$). The waveform presents only one peak and the first arrivals are impulsive.

Simple slopequakes were already presented under different names "slidequakes" (Gomberg et al., 2011), "Micro-earthquake" (Helmstetter and Garambois, 2010; Lacroix and Helmstetter, 2011), "quakes" (Tonnellier et al., 2013; Vouillamoz et al., 2017) or "landslide Micro-Quake (LMQ)" (Brückl, 2017). We here proposed the term "Slopequake" as the general name for these events. They are suspected to be associated to bounder or basal sliding (Helmstetter and Garambois, 2010; Gomberg et al., 2011; Walter et al., 2013b; Tonnellier et al., 2013) or fracturing of the slope (Helmstetter and Garambois, 2010).

Hybrid slopequakes are very similar to the events recorded on volcanoes and glaciers with the presence of fluids in conduits or crevasses (Chouet, 1988; Helmstetter et al., 2015b). The source of this event is assumed to be related to hydro-fracturing. The first high-frequency events corresponding to a brittle failure is followed by the flow of the water into the newly opened cracks (Chouet, 1988; Benson et al., 2008).

Presently, few studies have proposed inversion of the source tensor (Lévy et al., 2010). Therefore, the focal mechanism of the sources remain uncertainly known. Consequently, it remains undertermined if the Low-Frequency slopequakes are distant slopequakes (HF or Hybrid) or not. The lack of high frequencies may be explain either by attenuation during propagation of the seismic waves or by the source itself. Indeed, the rupture velocity may explain the difference of frequency content. Low-frequency earthquakes are generated on tectonic faults (Shelly et al., 2006; Brown et al., 2009; Thomas et al., 2016). They are characterized by low magnitude ($M_w < 2$) and short duration ($< 1$ s) and constitute at least part of the seismic tremor signal. Therefore, the main assumption for the source of these events are slow rupture (Thomas et al., 2016). LF-Slopequake may also be distant Mix-slopequake due to high attenuation due to highly fractured areas (Spillmann et al., 2007; Helmstetter and Garambois, 2010; Tonnellier et al., 2013). Finally, in glacier, low frequency icequakes dominated by surface waves are interpreted as surface sources generated by crevasse opening (Deichmann et al., 2000; Mikesell et al., 2012).



### 5.3.2 Complex Slopequake (CQ)

The second class of short duration signals has the same general properties than the Simple Slopequakes but exhibits particular frequency content or precursory events. These additional characteristics change the possible interpretation of the sources. Consequently, these signals are gathered in anoter class "Complex Slopequake". Three different sub-classes can be built:

– **Slopequake with harmonic coda (H-SQ)** (Figure 10): These signals are similar to the slopequake signals but present a monochromatic coda at high frequencies (i.e. 20 and 43Hz). The resonance is not present before the beginning of the signal and hence can not be due to anthropogenic noise (i.e. motors). In the case of Chamousset (Figure 10.b), the presence of this monochromatic coda is explained by the resonance of the rock column after the occurrence of the rock bridge breakage (Lévy et al., 2010). The resonant coda is rapidly attenuated with the distance and is not recorded by all
the sensors (Lévy et al., 2010). Considering the distance between the main scarp and the seismic arrays ($> 300$ m) at the Super-Sauze, similar resonant coda are observed at the end of certain rockfalls (Figure 4.d). The occurrence of this kind of resonance is very surprising in this case.

   – **Slopequake with precursors** (Figure 11): The third class of short duration signals are similar to the slopequake signals but are preceded by a precursory signal of smaller amplitude (Figure 11). The content of the precursory signal ranges
from 5 to 100 Hz depending on the site and is slightly lower than the highest frequency generated by slopequake-like event. The precursory arrival last up to 1.2 s in the presented examples and no clear phases are detected. The frequency content ranges from 5 to 100 Hz but varies significantly at each site. At all sites, the amplitude of the signal is significantly higher for one of the sensor (3 to 50 times higher) when considering vertical traces. The precursory signal is buried in the noise at the sensors with lowest amplitudes and the signal is similar to a LF-slopequake. Such events have never
been documented to our knowledge. They are likely to be generated by a strong and local source located at the very close vicinity of one of the sensor ($< 10$ m) due to the maximal amplitude ($> 10^5$ nm.s$^{-1}$) and the rapid decrease of the amplitude recorded by the other sensors. Although the signal is similar to certain earthquakes (the precursory signals interpreted as P-waves arrivals and the strong arrivals as surface waves), no earthquake location can explain the signal recorded at the time these events are recorded. Their occurrence in the night time also prevent any human activity to be
the source. The most probable source would then be the detachment of a single block and its fall in the vicinity to one of the sensor. This kind of precursory signals are observed for some rockfalls (Figure 4.a) and at a the Saint-Martin-le-Vinoux quarry (France; Helmstetter et al. (2011)). At the Saint-Martin-le-Vinoux underground quarry, the duration between the detachment and the signal impact is well correlated to the room height. This interpretation is coherent with the drop of amplitude before the more energetic event at the Chamousset rock column (Figure 11.c) where a progressive
decrease of the precursory signal is observed. However, on the other sites (Figure 11.a,.b) such decrease is not present. The one second lasting precursory signal has a constant amplitude and frequency content. Another interpretation could be that these precursory signals are a succession of overlaping slip or fracture events. The interpretation of these signals cannot be established with certainty and further analysis (i.e. location, time of occurrence) and other examples are needed to discriminate the mechanism at work.



### 5.3.3 Tremor-like slopequake

The last class of short duration signals often last between 1 and 5 seconds (Figure 12). They present a symmetrical waveform (S=50%) with emergent arrivals and slow decrease of the amplitude to the noise level. The frequency ranges from 5 Hz to 25 Hz. High-frequencies may be briefly recorded in certain events (Figure 12.c) . The maximal energy of the PSD corresponds to a frequency of 10 Hz while the mean energy corresponds to a frequency of 20 Hz. No seismic phases are identified. The signal is not recorded by all the sensors even when the sensors are organized in small arrays with short inter-sensor distances (< 50 m). Their waveforms and frequency content are similar to the one of the granular flows (Figure 5). Small debris flows have been observed at La Clapière and Super-Sauze landslides and are likely to generate seismic waves; however, small debris flows are not observed at the Pas de l'Ours landslide when these kinds of seismic signals are recorded. Another possible source mechanisms for such events may also be a very rapid succession (< 1 s) of shear events along the basal or the side bounding strike-slip faults (Hawthorne and Ampuero, 2017). Further investigations are needed to analyze their occurrences over time and their location to confirm one or the other assumptions.

## 6 Discussion

The proposed typology is summarized in Figure 13. The signals present significant differences with the chosen features. It must be noted that, in the field, the differentiation between flow and fall signals may be challenging. Indeed, some of the events are very likely a mix of these two sources. Rockfalls of various blocks may generate granular flows with metric block impacts, both overlapping in the recorded seismic signals. Presence of metric rocks is also observed in debris flow prone torrents; for this type of events, the block impacts within the mass flows are recorded in the seismic signals (Burtin et al., 2016).

Harmonic signals are recorded at different sites (Gomberg et al., 2011; Vouillamoz et al., 2017). These signals last from 1 to 5 s and repeat regularly during several minutes. The same signals are recorded at the La Clapière and the Aiguilles-Pas de l'Ours landslides with a fundamental frequency of $8 \pm 1$ Hz (Figure 14.b,c). At Séchilienne landslide, harmonic signals are also detected (Figure 14.d), mostly during the day, with different resonant frequencies between 2 and 12 Hz, simultaneously or for different time periods. Similar signals are observed at the Slumgullion and Super-Sauze but without clear harmonics in the PSD (Figure 14.e,f). In the case of Slumgullion (Figure 14.e), 90 repeaters of this event were measured during the 1 month observation period(Gomberg et al., 2011). The fundamental frequency is 11.9 Hz with a standard deviation of 0.7 Hz computed from the stack of the signals with a correlation coefficient higher than 0.7 (Gomberg et al., 2011). The authors argued that the resonance is a property of the source considering the stability of the fundamental frequency through time. They hypothesize that the waves were trapped along the side-bounding strike-slip fault generated by shear events. Resonance of fluid trapped in cracks or cavities is also a possible mechanism to generate these kind of signals (Benson et al., 2008; Tary et al., 2014a, b; Derode et al., 2015; Helmstetter et al., 2015b). It remains unclear whether natural cracks filled of water may generate this kind of signals on unstable slopes. Besides, the presence of pipes and drains on or in the vicinity of these sites is likely to explain the origin of these signals and their similarities. It justifies why these signals are not currently included in the present typology. They are likely not generated by a slope deformation process. More precise location and analyze of the temporal variations of





these events are needed to determine if they are actually generated by fluid resonance in fractures and must be integrated in the general typology.

The differences in the frequency content of simple slopequakes may be explained either by the attenuation of the high frequency at large distances during the propagation or by different rupture velocity and/or the presence of fluid in the fault plane. It is currently impossible to distinguish these two effects as the source time function cannot be inverted. Simple slopequakes are currently assumed to be generated by shear movement along a plane or tensile opening of cracks (Spillmann et al., 2007; Helmstetter and Garambois, 2010; Gomberg et al., 2011). At the Chamousset rock column, the source mechanism is retrieved by the P -and S-waves amplitude ratio (Lévy et al., 2010). Shear events are found to be located at the bottom of the column while tensile opening is occurring in the upper part (Lévy et al., 2010). To the best of our knowledge, for soft-rock landslides, no source mechanism was modeled. For fine material, the inversion of the source mechanism is currently challenging due to: 1) the attenuation of the seismic waves and especially of the first arrivals, 2) the inaccurate location, in particular the depth of the source, 3) the complexity of the landslide geometry making several source mechanisms possible at the same location. Moreover, the limited number of installed sensors and the geometry of the networks (controlled by the location of the stable zones; (Spillmann et al., 2007)) is not always optimal to compute source focal mechanisms.

No long-lasting tremors are presented in this study. Schöpa et al. (2017) recorded a tremor with gliding before the occurrence of the Askja caldera landslide. Similar tremors are found on the Whillans ice stream in Antarctica during slow slip events (Paul Winberry et al., 2013; Lipovsky and Dunham, 2016), which repeat twice a day with a slip of about 10 cm lasting for about 20 mn. Therefore, such signals may also occur during the nucleation phase of landslide failure. The question remains unclear if they are not observed because landslide acceleration is aseismic due to high pore fluid pressure (Scholz, 1998) or low normal stress at the sub-surface of the slope.

Thanks to the catalogs of endogenous seismic events being progressively built, solid assumptions on the nature of several seismic sources can be proposed. However, difficulties still arise in providing an exhaustive description of all the sources, particularly those generating short-duration signals. Several limitations currently prevent such analysis. First, the location of the sources remain difficult to establish due to the complexity of some of the signals, the size of the instrumented sites and the geometry (number, location) of the sensors installed close to the unstable slopes. In order to improve the precision of the location, realistic 3-D models in both P- and S-waves are needed (Spillmann et al., 2007) as well as appropriate location strategies taking into account the complexity of the signal phases. Several approaches have been proposed based either on the amplitude of the signal (Battaglia and Aki, 2003; Walter et al., 2017), on the surface waves correlation (Lacroix and Helmstetter, 2011; Gomberg et al., 2011; Tonnellier et al., 2013), on the picking of the first arrivals (i.e. P-waves, (Spillmann et al., 2007; Lévy et al., 2010; Vouillamoz et al., 2017) or the picking of surface waves, (Hibert et al., 2014a; Levy et al., 2015). The location of the epicenter of most of the events seems coherent with the instabilities deformation although resolving dispersion and 3-D heterogeneities of the velocity fields currently prevents to infer the depth of the events and their focal mechanisms. Most of the seismic networks are also not dense enough to resolve both location at depth and focal mechanisms.

A complementary approach to explain the origin of the sources is the analysis of their occurrence with respect to surface or basal displacement and monitoring of the water content and pore fluid pressures. It requires both exhaustive catalogs of





landslide seismicity over long time periods and continuous and distributed datasets of displacements and pore fluid pressures. Automatic classification algorithms have shown their efficiency to classify landslide seismic signals (Hammer et al., 2012; Dammeier et al., 2016; Provost et al., 2017; Maggi et al., 2017; Hibert et al., 2017b). Template-matching filters have been used in many studies of landslides and glaciers (Allstadt and Malone, 2014; Yamada et al., 2016a; Poli, 2017; Helmstetter et al.,

2015a, b; Bièvre et al., 2017; Helmstetter et al., 2017a) in order to detect and classify seismic signals. This method allows to detect and classify automatically seismic signals by scanning continuous data to search for signals with waveforms similar to template signals. It can detect seismic signals of very small amplitude, smaller than the noise level. Seismic signals are grouped in clusters of similar waveforms, implying similar source locations and focal mechanism.

    In addition to the characteristics of seismic signals, further information on the sources processes can be obtained from the

distribution of the events in time, space and size. Events that occur regularly in time with similar amplitudes are likely associated with the repeated failure of an asperity surrounded by aseismic slip, for instance, at the base of a glacier (Helmstetter et al., 2015a) or of a landslide (Yamada et al., 2016a; Poli, 2017; Helmstetter et al., 2017a). Seismic signals of repeating slope quakes on landslides are usually emergent and low frequency (mean frequency of about 5 Hz), but this could be due to attenuation, because the source-sensor distance in these studies is 1 km or longer. Signal amplitudes and recurrence times often display

progressive variations in time. In contrast, events that are clustered in time and space, with a broad distribution of energies, are more likely associated with the propagation of a fracture (Helmstetter et al., 2015b). The daily distribution of events time can also be helpful to identify anthropic sources, that occur mostly during the day. In contrast, natural events are more frequently detected at night, when the noise level is smaller.

    Simulations and models are also required to explain the current observations. Indeed, experimental results suggest an in-

crease of acoustic emissions correlated with the increase of the slope velocity (Smith et al., 2017) or an increase of acoustic emission due to the creation of the rupture area (Lockner et al., 1991). Acceleration of pre-existing rupture surface(s) seem to be the mechanism responsible for the seismicity recorded before large rockslide collapse. Yamada et al. (2016a); Poli (2017); Helmstetter et al. (2017a) argued that the high correlation between the repetitive events could only be explained by stick-slip movement of the locked section(s), while a cracking process would imply a migration of the location of the events and a change

in the events waveforms. Schöpa et al. (2017) argued that the presence of gliding frequencies could only be produced by similar sources and hence close location. On the contrary, in the case of the Mesnil-val column, Senfaute et al. (2009) interpreted the evolution from high frequency to low frequency events as the progressive formation of the rupture surface followed by the final rupture process immediately before the column collapse where both tensile cracks and shearing motion on the created rupture are generated. However, most collapses occurred without precursory sequences (Allstadt et al., 2017) and during long-term

monitoring most of the deformation seems to occur aseismically (Lacroix and Helmstetter, 2011). Most of the studies have shown that the number of events is significantly correlated with rainfall and displacement rates (Amitrano et al., 2005; Helmstetter and Garambois, 2010; Walter et al., 2012; Brückl et al., 2013; Tonnellier et al., 2013; Vouillamoz et al., 2017) although some increases of seismicity rates are not correlated to any surface displacement (Helmstetter and Garambois, 2010; Walter et al., 2012; Tonnellier et al., 2013; Vouillamoz et al., 2017). Further investigations and experimentation are needed to quantify

this relationship. Experimental and numerical simulations are needed to better characterize the seismic signals associated to



the slow motion and fast acceleration toward failure of unstable slopes. Moreover, the condition of occurrence of precursory seismic events during the nucleation phase of the landslide failure must be also better understand.

# 7   Conclusions

Over the last decades, numerous studies have recorded seismic signals generated by various types of landslides (i.e. slide, topple, fall and flow), for different kinematic regimes and rock/soil media. These studies demonstrated the added-value of analyzing landslide-induced micro-seismicity to improve our understanding of the mechanisms and to progress in the forecast of landslide evolution.

In this work we propose a review of the endogenous seismic sources generated by the deformation of unstable slopes. A dataset of fourteen slopes is gathered and analyzed. Each of the source is described by nine quantitative features of the recorded seismic signals. Those features provide distinct characteristics for each type of source. A library of relevant signals recorded at relevant site is shared as supplementary material. We propose three main class "slopequake", "rockfall" and "granular flow" to describe the main type of deformation observed on the slopes. Slopequakes are related to shearing or fracturing processes. This family exhibits the most variability due to the complexity of the sources. These variations are likely to be generated by different source mechanisms. "Rockfall" and "granular flow" classes are associated to mass propagation on the slope surface. They are distinguishable by the number of peaks clearly identified in the seismic signals.

Presently, several descriptions of the seismic sources are proposed for each study case. We believe that a standard typology will allow to discuss and compare seismic signals recorded at many unstable slopes. We encourage future studies to use and possibly enrich the proposed typology. This also requires publication of the datasets and/or catalogs to progress towards a common interpretation. Recently, organizations such as the United States Geological Survey (USGS) or the French Landslide Observatory (OMIV) have started this work (RESIF/OMIV, 2015; Allstadt et al., 2017).

A better understanding of the different sources endogenous to unstable slopes can also be achieved through the development of new adapted processing strategies to classify, locate and invert focal mechanism. Those developments must also be associated with the deployment of denser seismic networks, by taking advantages of the recent arrival on the market of relatively cheap and autonomous seismometers (eg. ZLand node systems, RasberryShake systems). Moreover, the recent operational applications of Ground-Based SAR (Synthetic Aperture Radar) and terrestrial LiDAR technologies for monitoring purposes shows their relevance to monitor distributed surface displacements. On-going monitoring on several landslides combining those innovative approaches will certainly help to associate SQ events to deformation processes.

The proposed typology will help to constrain the design of new models to confirm the assumptions on the nature and the properties of the seismic sources. This will be particularly important for 1) explaining the variability of the SQ sources observed at the sites, 2) progressing in the physical understanding of the SQ sources, and 3) ascertaining the spatio-temporal variations of the seismic activity observed at some unstable slopes in relation with their deformation as well as, with external forcings such as intense rainfalls and earthquakes.



*Data availability.* The library of the endogenous seismic signals recorded at the sites and described in the manuscript is shared as supplementary material. The seismic data are shared in Matlab *.mat* format.

*Acknowledgements.* This work was carried with the support of the French National Research Agency (ANR) through the projects HYDROSLIDE "Hydrogeophysical Monitoring of Clayey Landslides", SAMCO "Society Adaptation to Mountain Gravitational Hazards in a

5    Global change Context", and TIMES "High-performance Processing Techniques for Mapping and Monitoring Environmental Changes from Massive, Heterogeneous and High Frequency Data Times Series". Additional support by the Open Partial Agreement "Major Hazards" of Council of Europe through the project "Development of Cost-effective Ground-based and Remote Monitoring Systems for Detecting Landslide Initiation" was available. The continuous seismic data were provided by the Observatoire Multi-disciplinaire des Instabilités de Versant (OMIV) (RESIF/OMIV, 2015). Some seismic signals analyzed were acquired with seismometers belonging to the French national pool of

10   portable seismic instruments SISMOB-RESIF. The authors thank J. Gomberg for the access to the data of the Slumgullion slope, constructive discussions and review of the early version of this paper as well as Nick Rosser and Emma Vann Jones for the access to the data of the North Yorkshire cliff. The author also thank Naomi Vouillamoz (University of Stuttgart) and Pascal Diot (ONF-RTM) for helping in the data acquisition at, respectively, the Pechgraben site and the Aiguilles-Pas de l'Ours site.




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



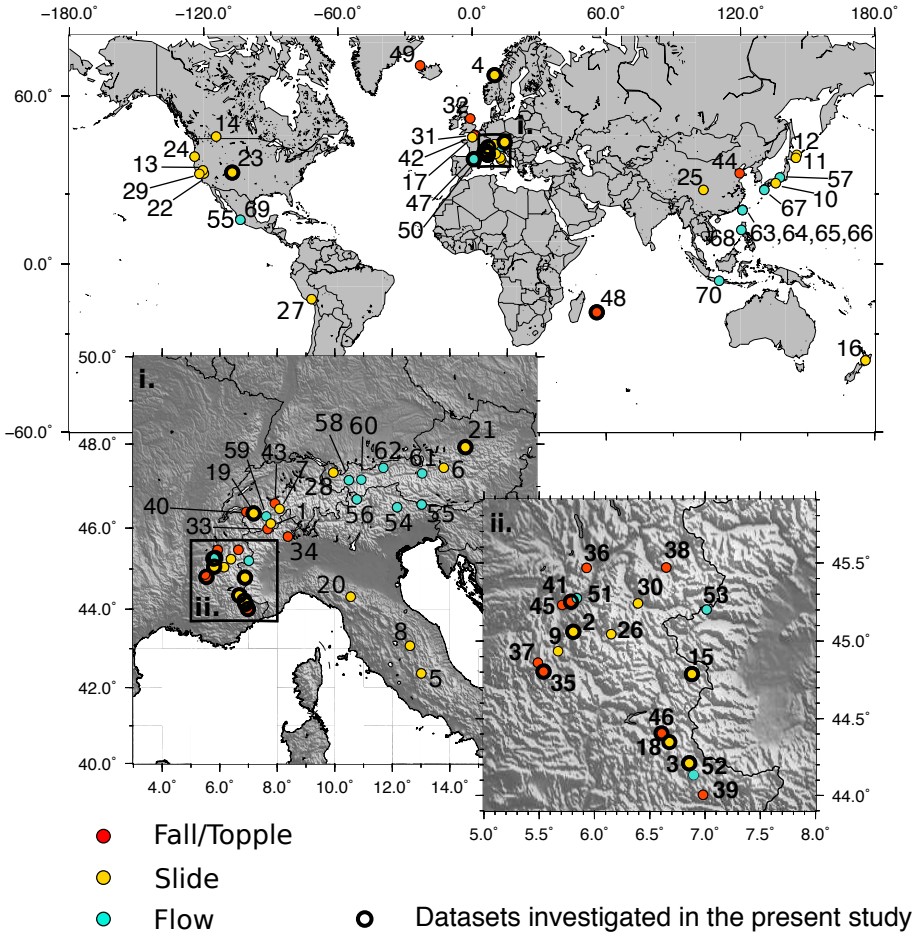

**Figure 1.** Map of seismologically-instrumented unstable slopes. Refer to table 1 for the name of the sites corresponding to the number.

Table 1: Table of the instrumented sites. The bolded names correspond to the sites investigated in the present paper to establish the typology.

| Number | Site | Location | Type | Material | Sensor | Duration | Reference/Research Group |
|---|---|---|---|---|---|---|---|
| 1 | Randa | Switzerland | Slide | Hard rock | G | SC | Spillmann et al. (2007) |
| **2** | **Séchilienne** | France | Slide | Hard rock | G, SP | P | RESIF/OMIV (2015); Helmstetter and Garambois (2010); Lacroix and Helmstetter (2011) |
| **3** | **La Clapière** | France | Slide | Hard rock | SP(?) | P | RESIF/OMIV (2015); Palis et al. (2017) |
| **4** | **Aaknes** | Norway | Slide | Hard rock | G,BB | P | Roth et al. (2008) |
| 5 | Peschiera Spring | Italy | Slide | Hard rock | A | SC | Lenti et al. (2013) |
| 6 | Gradenbach | Austria | Slide | Hard rock | SP | P(?) | Brückl et al. (2013) |
| 7 | Alestch-Moosfluh | Switzerland | Slide | Hard rock | BB | P | Helmstetter et al. (2017b) |
| 8 | Torgiovannetto, Assise | Italy | Slide | Hard rock | SP | SC | Lotti et al. (2015) |
| 9 | Harmalière | France | Slide | Hard rock | SP,BB | P | Bièvre et al. (2017) |
| 10 | Akatami landslide | Japan | Slide | Hard rock | (?) | (?) | - |
| 11 | Akkeshi landslide | Japan | Slide | Hard rock | SP | P | Doi et al. (2015) |
| 12 | Rausu landslide | Japan | Slide | Hard rock | BB | P | Yamada et al. (2016a) |
| 13 | Fergurson slide / Mercel River | USA / California | Slide | Hard rock | (?) | (?) | Harp et al. (2008) |
| 14 | Turtle Mountain - Frank slide | Canada | Slide | Hard rock | G | P | Chen et al. (2005) |
| **15** | **Aiguilles** | France | Slide | Soft rock / Earth | BB | SC | RESIF/OMIV (2015) |
| 16 | Utiku | New Zealand | Slide | Soft rock / Earth | (?) | P | Voisin et al. (2013) |

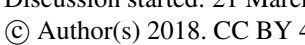


| Number | Site | Location | Type | Material | Sensor | Duration | Reference/Research Group |
|---|---|---|---|---|---|---|---|
| 17 | Villerville | France | Slide | Soft rock / Mud | BB | SC,P | RESIF/OMIV (2015) |
| **18** | **Super-Sauze** | France | Slide | Soft rock / Mud | SP | P, RC | RESIF/OMIV (2015); Walter et al. (2012); Tonnellier et al. (2013); Vouillamoz et al. (2017) |
| **19** | **Pont Bourquin** | Switzerland | Slide | Mud | SP(?) | P | Mainsant et al. (2012a); Larose et al. (2015) |
| 20 | Valoria | Italy | Slide | Mud | SP | SC | Tonnellier et al. (2013) |
| **21** | **Pechgraben** | Austria | Slide | Mud | SP,BB | RC | Vouillamoz et al. (2017) |
| 22 | US highway 50, CA | USA | Slide | Earth | G | P | USGS |
| **23** | **Slumgullion** | USA | Slide | Earth | G | RC | Gomberg et al. (1995, 2011) |
| 24 | Millcoma Meander, Oregon | USA | Slide | Earth | G | P | USGS |
| 25 | Xishancun | China | Slide | Earth | BB | SC | - |
| 26 | Chambon Tunnel | France | Slide | Earth | SP | P | - |
| 27 | Maca | Peru | Slide | Soft rock / Earth | SP | P(?) | Larose et al. (2017) |
| 28 | Heumoes | Germany | Slide | Soft rock / Earth | SP | RC | Walter et al. (2011) |
| 29 | Mission Peak landslide | USA / California | Slide | Soft rock / Earth | BB | P | Hartzell et al. (2017) |
| 30 | Char d'Osset | France | Slide, Fall | Soft rock / Mud | | | |
| 31 | Mesnil-Val | France | Fall | Hard rock | G | SC | Amitrano et al. (2005); Senfaute et al. (2009) |
| 32 | North Yorkshire coast | United Kingdom | Fall | Hard rock | BB | P | Norman et al. (2013) |
| 33 | Matterhorn peak/Mont Cervin | Italy | Fall | Hard rock | G | RC | Amitrano et al. (2010) |
| 34 | Madonna del sasso | Italy | Fall | Hard rock | SP | P(?) | Colombero et al. (2018) |
| **35** | **Chamousset** | France | Fall | Hard rock | SH | RC | Lévy et al. (2010); Bottelin et al. (2013b) |
| 36 | Mont-Granier | France | Fall | Hard rock | BB | P | - |
| 37 | Les Arches | France | Fall | Hard rock | SP | P(?) | Bottelin et al. (2013a, b) |
| 38 | La Praz | France | Fall | Hard rock | SP | P(?) | Bottelin et al. (2013b) |
| 39 | Rubi | France | Fall | Hard rock | SP | P(?) | Bottelin et al. (2013b) |
| 40 | La Suche | Switzerland | Fall | Hard rock | SP | P(?) | Bottelin et al. (2013b) |
| **41** | **St. Eynard cliff** | France | Fall | Hard rock | SP | P(?) | - |
| 42 | Cap d'Ailly | France | Fall | Hard rock | | | - |
| 43 | Lauterbrunnen valley | Switzerland | Fall | Hard rock | BB | SC | Dietze et al. (2017a, b) |
| 44 | Three Brothers | USA | Fall | Hard rock | SP | SC | Zimmer and Sitar (2015) |
| 45 | Mount Néron | France | Fall (triggered) | Hard rock | BB | SC | Bottelin et al. (2014) |
| **46** | **Riou Bourdoux** | France | Fall (triggered) | Hard rock | SP,BB | SC | Hibert et al. (2017a) |
| 47 | Montserrat | Spain | Fall (triggered) | Hard rock | SP | SC | Vilajosana et al. (2008) |
| **48** | **Piton de la Fournaise caldeira** | France | Fall, Flow | Volcanic rock | BB | P | OPVF/IPGP, Hibert et al. (2011, 2014a); Levy et al. (2015); Hibert et al. (2017c) |
| 49 | Bolungavík - Oshlíðslope | Iceland | Fall, Flow | Hard rock | A | P | Bessason et al. (2007) |
| **50** | **Rebaixader** | Spain | Flow | Debris | G | P | Abancó et al. (2012, 2014); Hürlimann et al. (2014); Arattano et al. (2014) |
| 51 | Manival torrent | France | Flow | Debris | G | P | Navratil et al. (2012) |
| 52 | Réal torrent | France | Flow | Debris | G | P | Navratil et al. (2012) |
| 53 | Marderello torrent | Italy | Flow | Debris | G | P | Arattano et al. (2016) |
| 54 | Acquabona torrent | Italy | Flow | Debris | G | P(?) | Berti et al. (2000); Galgaro et al. (2005) |
| 55 | Moscardo torrent | Italy | Flow | Debris | SP | P | Arattano and Moia (1999) |
| 56 | Gadria torrent | Italy | Flow | Debris | G | P | Arattano et al. (2016) |
| 57 | Mt. Yakedake volcano - Kamikamihorizawa Creek | Japan | Flow | Debris | | | Suwa et al. (2009) |
| 58 | Lattenbach torrent | Austria | Flow | Debris | G | P(?) | Schimmel and Hübl (2016); Kogelnig et al. (2014) |
| 59 | Illgraben torrent | Switzerland | Flow | Debris | G | P | Burtin et al. (2014); Walter et al. (2017) |
| 60 | Farstrine torrent | Austria | Flow | Debris | G | P(?) | Schimmel and Hübl (2016) |
| 61 | Wartschenbach torrent | Austria | Flow | Debris | G | P(?) | Schimmel and Hübl (2016) |
| 62 | Dristenau torrent | Austria | Flow | Debris | G | P(?) | Schimmel and Hübl (2016) |
| 63 | Shenmu creek | Taiwan | Flow | Debris | G | P | Yin et al. (2011) |
| 64 | Ai-Yu-Zi creek | Taiwan | Flow | Debris | G | P | Huang et al. (2007) |
| 65 | Fong-Ciou creek | Taiwan | Flow | Debris | G | P | Huang et al. (2007) |
| 66 | Chenyoulan creek | Taiwan | Flow | Debris | G | SC | Burtin et al. (2013) |
| 67 | Mt. Sakurajima Volcano - Nojiri Torrent | Japan | Flow | Debris | G | P | Itakura et al. (2000) |
| 68 | Mount Pinatubo | Philippines | Flow | Debris | G | P | Marcial et al. (1996) |
| 69 | La Colima volcano | Mexico | Flow | Debris | LP | P | Zobin et al. (2009); Vázquez et al. (2016) |
| 70 | Merapi volcano flanks | Indonesia | Flow | Debris | G | P | Lavigne et al. (2000) |

G: Geophone (f = [0.1-10] kHz); SP: Short-Period (f = [0.1-100] Hz); BB: Broad-Band (f = $[10^{-2}$-100] Hz); A: Accelerometer;

P: Permanent monitoring; RC: Repetitive Campaigns; SC: Single Campaign.

OPVF/IPGP: Volcanological Observatory of the Piton de la Fournaise / Institut de Physique du Globe de Paris.

USGS: United States Geological Survey.



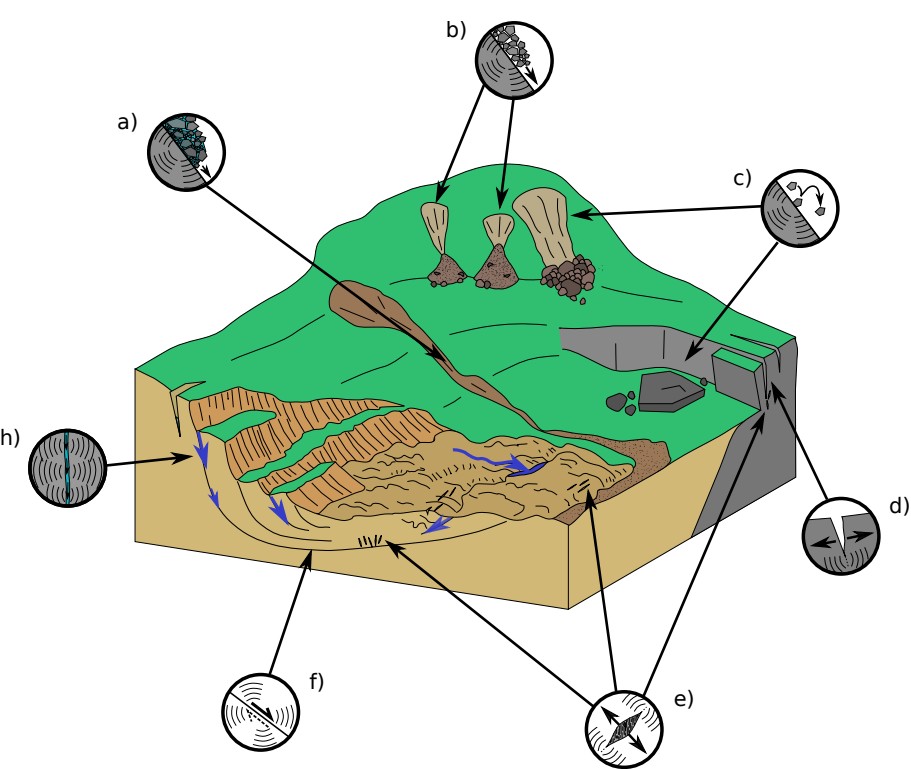

**Figure 2.** Conceptual scheme of the landslide endogenous seismic sources with a) wet granular flow, b) dry granular flow, c) rockfall, d) tensile fracture opening, e) tensile cracks opening, f) shearing and h) fluid migration in fracture.



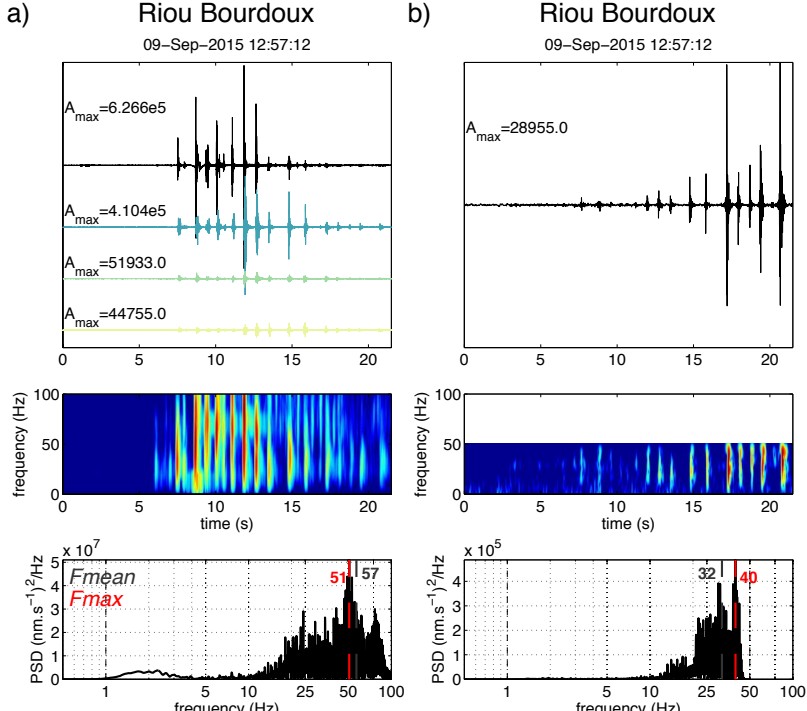

**Figure 3.** Example of one controlled rockfall (mass= 430kg) at the Riou-Bourdoux catchment (Hibert et al., 2017a) recorded by SP seismometer located at 50 m of the rock departure (left) and recorded by BB seismometer near the rock arrival (right). The waveforms of the vertical traces are plotted on the upper part of the figure. The amplitude are normalized on the trace with the maximal amplitude (black). The maximal amplitudes of all the traces are plotted on the sub-plot. The spectrogram is plotted on the middle part of the figure and normalized to the maximal energy. The lower part of the figure represents the PSD of the most energetic trace and the frequency corresponding to the maximum and the mean of the PSD are plotted in red and gray respectively.




**Figure 4.** Rockfall events recorded at a) and d) Super-Sauze (France), b) at the Séchilienne (France), c) Chamousset, e) Aaknes and f) Mount Saint-Eynard slopes. See Figure 3 for description of the figure.





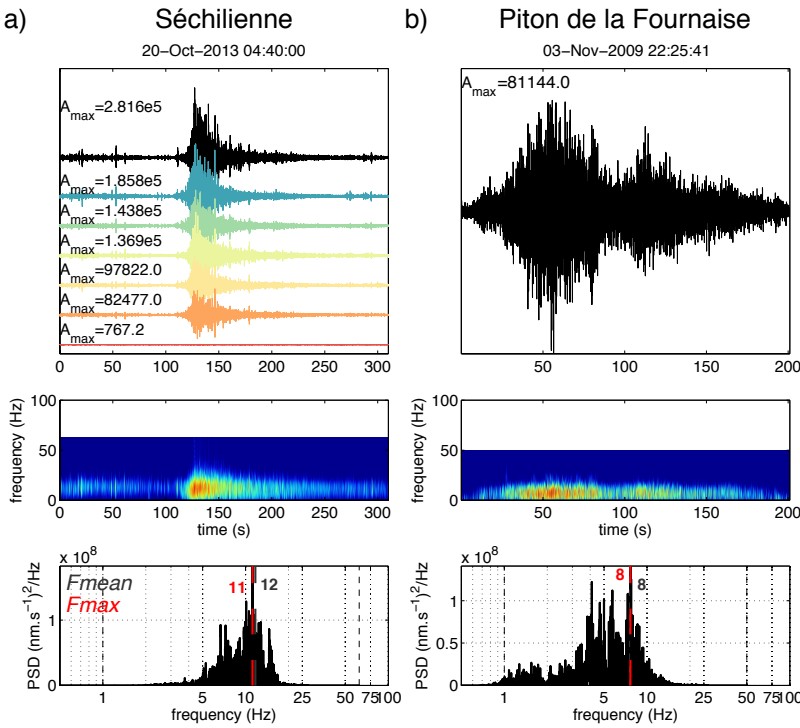

**Figure 5.** Dry granular flow events recorded at a) Séchilienne and b) the Piton de la Fournaise Caldera. See Figure 3 for description of the figure.



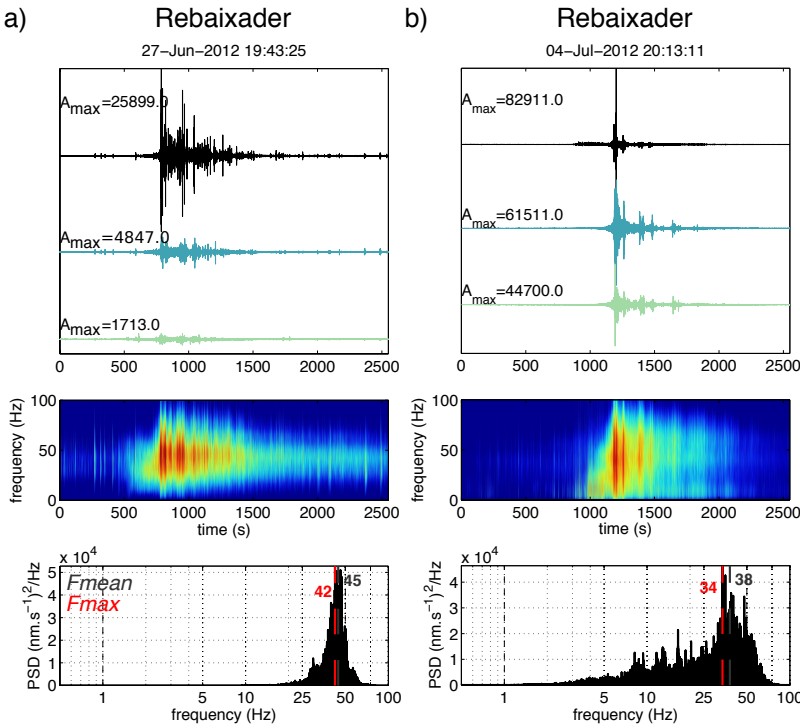

**Figure 6.** Wet granular flow events recorded at Rebaixader torrent (Abancó et al., 2012; Hürlimann et al., 2014; Arattano et al., 2016). See Figure 3 for description of the figure.




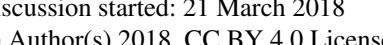

**Figure 7.** Low-Frequency Slopequakes recorded at the a) Slumgullion, b) Pont-Bourquin, c) La Clapière and d) Aiguilles-Pas de l'Ours slopes. See Figure 3 for description of the figure.



**Figure 8.** High-Frequency Slopequakes recorded at the a) Super-Sauze, b) Séchilienne, c) Pont-Bourquin, d) La Clapière, e) Aaknes, and f) Slumgullion slopes. See Figure 3 for description of the figure.





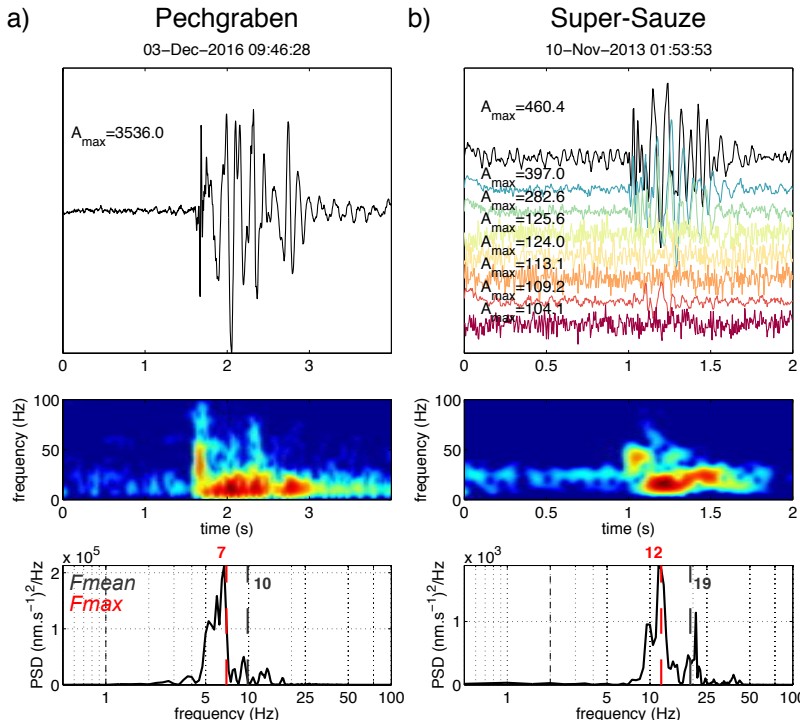

**Figure 9.** Mix-Slopequake recorded at the a) Pechgraben and b) Super-Sauze landslide. See Figure 3 for description of the figure.





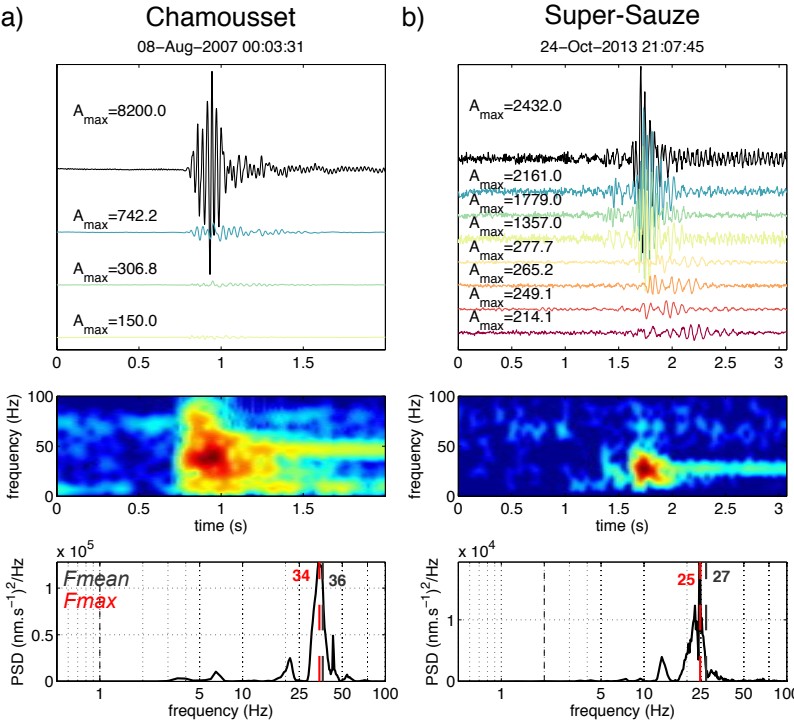

**Figure 10.** Examples of slopequakes with resonance in the coda recorded at a) Chamousset and b) Super-Sauze slopes. See Figure 3 for description of the figure.



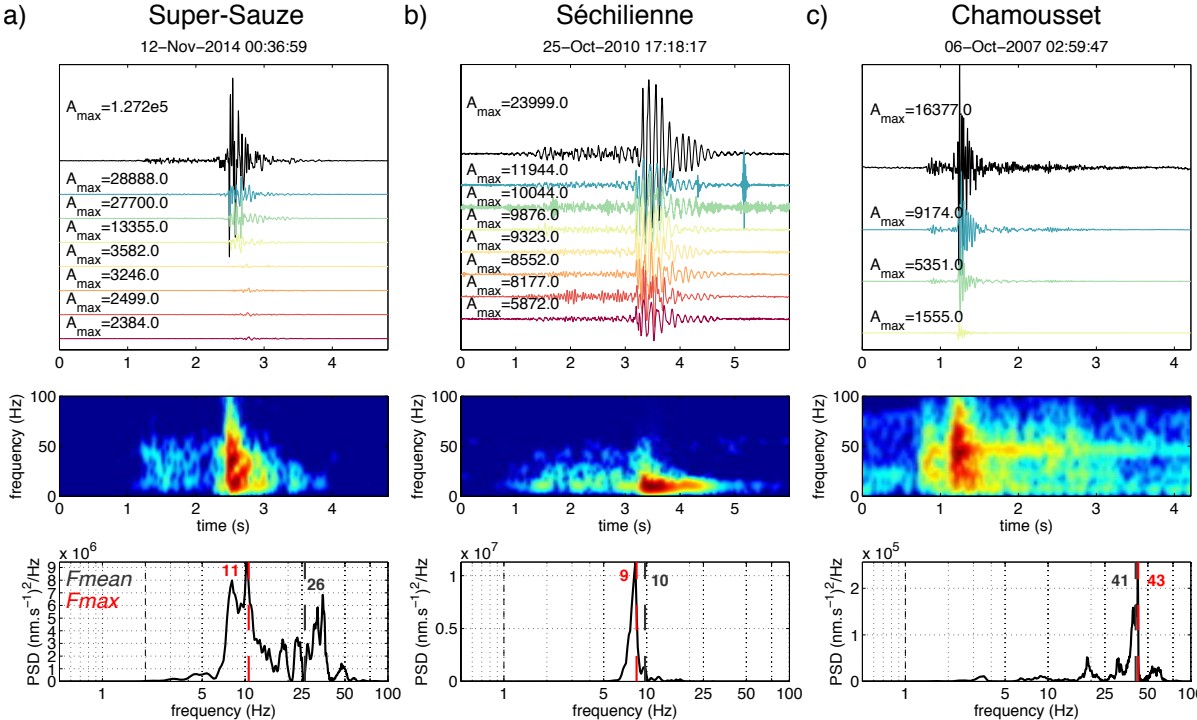

**Figure 11.** Examples of Slopequakes recorded at the a) Super-Sauze, b) Séchilienne and c) Chamousset slopes. See Figure 3 for description of the figure.



**Figure 12.** Examples of repetitive Slopequakes recorded at the a),c) Super-Sauze, b) La Clapière and d) Aiguilles-Pas de l'Ours slopes. See Figure 3 for description of the figure.

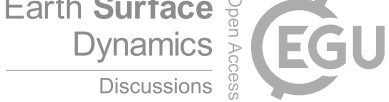

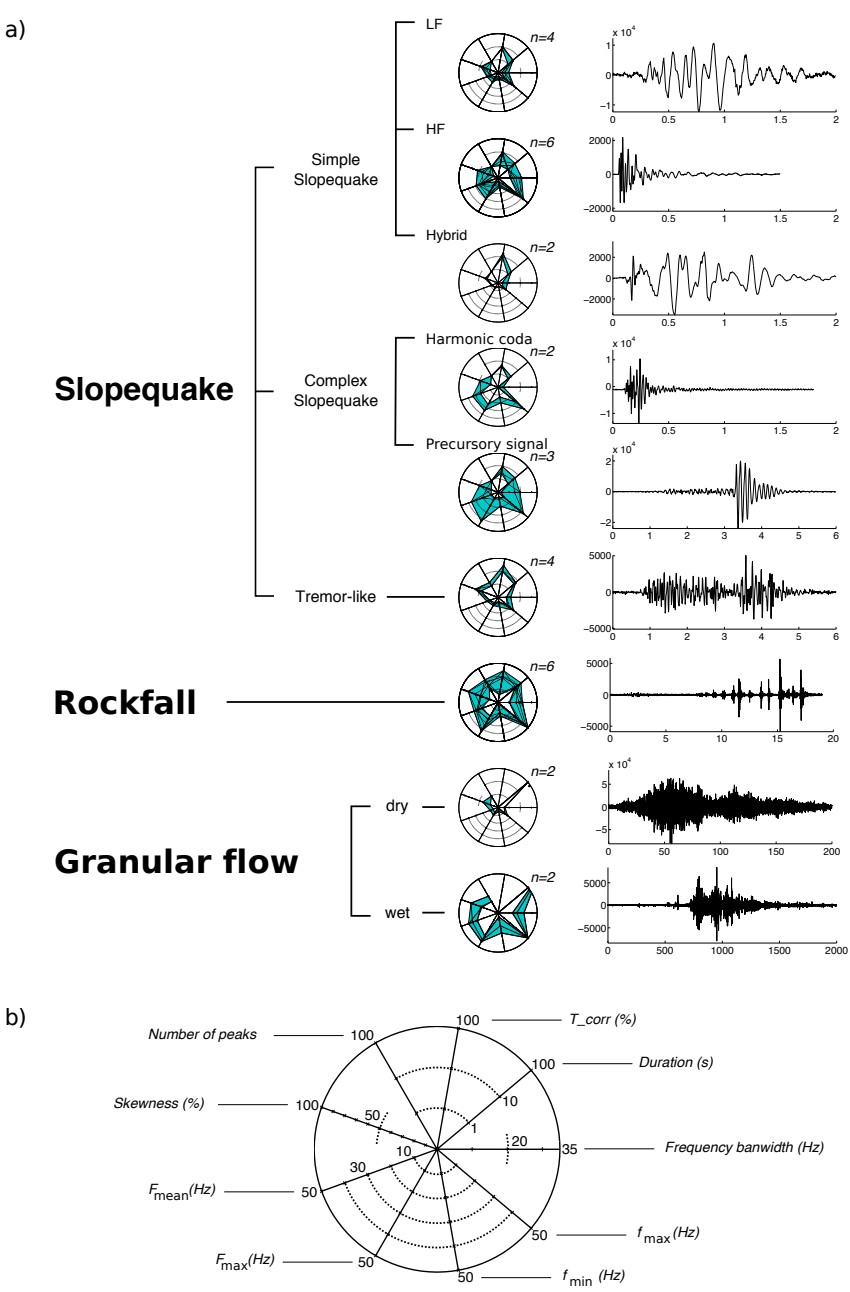

**Figure 13.** a) Summary of the proposed classification with plot of the attributes for the examples presented in the precedent figures and an example of waveform for each class. The number of examples plotted per attribute plot (n) is indicated and the variability of the shapes is indicated in blue. The convention for the attribute plot is presented in b).





**Figure 14.** Examples of pure harmonic signals recorded at the a) Pechgraben, b) La Clapière and c) Aiguilles-Pas de l'Ours, d) Séchilienne, e) Slumgullion and f) Super-Sauze slopes. See Figure 3 for description of the figure.