# Peer review of "Towards a standard typology of endogenous landslide seismic sources"

_Earth Surface Dynamics, 2018_

## Referee Comment (RC1) · V. Coviello (Referee) · 6 Apr 2018

The paper by Provost et al. proposes a classification of landslide-induced seismic signals detected at close distances (< 1 km) based on data gathered in 14 field sites. A standard classification of seismic sources generated by mass movements detected at local scale is definitively of interest for the scientific community dealing with environmental seismology. However, it is not clear if this paper is a research or review article. A collection of study sites and already published seismic data is presented but a systematic testing of the proposed classification is missing. I would suggest the authors to choose one of these two possible formats, research or review article, and to consequently reshape the manuscript. I recommend the paper for publication in Earth Surf. Dynam. after major revisions.

[Figure]

Major comments:

I think that the abstract needs a significant rewording. The first lines sound like an intro-duction on environmental seismology. Please focus more on objectives, methods and results of your work. In addition, I disagree with the statement "The seismic networks installed on these sites are roughly similar (i.e. sensor, network geometry)". What does "roughly similar" mean? There is a significant difference between a BB seismic network installed on a large, slow moving earth-flow and a linear array of geophones deployed along a debris flow channel. Moreover, if the authors are focusing on "seismic events detected at close distances (< 1 km)" the sensor network characteristics and geometry, as well as the geological and geomorphological contexts, have a strong impact on the recorded signals.

To achieve a standard source characterization, in my opinion there are three major top-ics that would need to be addressed: i) distance sensor-source, ii) typology of sensor, iii) sensor installation methods. Given the pretty ambitious title, I would expect some discussion of their effects on landslide sources. However, I have the impression that the paper leaves more open questions than clear responses. In the following more details on how these three aspects have not been adequately addressed are given.

i) The authors briefly touch this point in the discussion: "The differences in the fre-quency content of simple slopequakes may be explained either by the attenuation of the high frequency at large distances during the propagation or by different rupture velocity and/or the presence of fluid in the fault plane". I encourage them to stress more on the possible limitations of a spectral analysis to be employed in a general classification. For instance, consider what was already published about the effect of the sensor-source distance on the seismic signal produced by flow processes (Gimbert et al., 2014; Schmandt et al., 2013). ii) At P7 L5-6 the authors state "The relatively low energy released by the landslide related sources makes the choice of the seismologi-cal instruments to deploy very important". I agree, and I think that this point should be developed more. Section 3.1 describes the main classes of sensors employed for the

detection of mass movements but I do not see a proper discussion of this point when the authors present their dataset. iii) Considering flow detection at channel scale, the sensor installation method has a strong impact on the features of the recorded signal, both in amplitude (e.g., Coviello et al., 2015) and frequency domain (e.g., supplementary material of Kean et al., 2015). Again, this issue is shortly introduced at P3 L11 "The location of the sensors and the type of waveguide are also critical to capture the slope behavior" but a discussion based on the analyzed dataset is missing.

Standardized datasets and field experiments are probably needed to systematically address those topics. I am skeptical about the possibility to develop a standardized source-mechanisms characterization of landslide-induced seismic signals from a collection of heterogeneous case studies.

Additional comments:

P2 L5: references needed

P2 from L26: concerning repeaters, I would suggest to the authors to read the reviews of the paper by SchoÌĹpa et al. (2017), an interesting discussion is made there on this point

P3 L16: "low frequency ranges (1-500 Hz)", why do you define this pretty broad frequency range as low? Compared to what?

P4 L30: "13 monitored sites", 13 or 14?

P4 L33-35: concerning "we first discuss all the physical processes that occur on landslides... We further present the seismologically-instrumented landslides in the world... Then we establish a classification scheme", I suggest the authors to rephrase in order to be more realistic. I think that the main physical processes were discussed, that only a few (14) of the seismologically-instrumented landslides in the world were presented and that a possible classification scheme was proposed.

P8 L13-24: these lines sound more as part of introduction than data. The paragraph

**ESurfD**
data should start from current L25 but a description of the analyzed dataset is actually missing: which is the length of the analyzed time series? How many events did you analyze? How did you select the events analyzed in the following paragraphs? I guess those were well-known events, or did you applied an automatic detection methods?

P8 L26: "For all sites, the instruments are deployed close to the landslide", what does "close" mean? Please be more specific. I guess that authors agree that, for example, two seismic sensors, one installed at 10 m and another one at 900 m from the very same landslide, would record signals pretty different, especially in terms of spectral content.

P15 L14: "The signals present significant differences with the chosen features", please reword, the reader does not understand the meaning of this sentence.

P15 L15: "in the field, the differentiation..." I am not sure to have correctly understood, maybe you meant "only from the seismic signal analysis, the difference between..."?

P17 L23: please avoid references that are not published work, i.e. Helmstetter et al. (2017a), especially if the reference is used to support a very strong statement such as "the high correlation between the repetitive events could only be explained by stick-slip movement of the locked section(s)". A sentence like this must be accompanied by supporting data or published results.

P17 L29: concerning "most collapses occurred without precursory sequences (Allstadt et al., 2017)", I would suggest tuning down this statement, which is also in contradiction with P2 L24. There are a number of cases where precursory seismic signals related to small rockfalls were documented, especially when a station is installed nearby the slope or there is a local monitoring network. On the contrary, when the closest station is distant or we do not dispose of other monitoring data, recognizing those precursory events is difficult but potentially there are. I also believe that the reference Allstadt et al. (2017) is not consistent here

P18 L16-20: I do agree with "several descriptions of the seismic sources are proposed for each study case" and a standard classification would help to discuss and compare landslide-induced seismic signals. I understand that the authors are proposing their classification as general reference, but I would suggest to the authors to delete the sentence "we encourage future studies to use and possibly enrich the proposed ty-pology". In my opinion the scientific community does not need to be encouraged to adopt one classification or another. By the way, why you do not adopt the classification proposed by Allstadt et al. (2017)?

P18 L25-27: reference needed

Reference list: the style in not consistent with the journal guidelines, in many cases the doi is missing, there are repeated references (Hibert et al., 2014a), others are missing (Provost et al., 2018) and there is some text here and there probably out of place (e.g., P29 L10-11). An accurate revision of the reference list is needed. Moreover, I do acknowledge the significant contribution of some of the authors to the field but I have the impression that self-citations are really abundant (five papers by Hibert et al., six by Helmstetter et al.). Please try to select your most significant works and refer to them.

Figure 1: I would prefer the author to focus more on the sites from which they present some data instead of showing a collection of points in a global map. In addition, Figure 1 is redundant if one considers the list presented in Table 1

Table 1: some details/revisions are needed. 7 Alestch-Moosfluh: this landslide is also monitored with a geophone network (Manconi and Coviello, 2018); 8 Torgiovannetto, Assise: please modify in Assisi; 15 Aiguilles: Aiguilles Pas de l'Ours?; 22 US high-way 50, CA: there is no reference/website about that?; 24 Millcoma Meander, Oregon: same as above; 33 Matterhorn peak/Mont Cervin: please use the international name (Matterhorn) or the Italian one (Cervino) and add the reference describing the more recent monitoring network (Occhiena et al., 2012); 48 Piton de la Fournaise caldeira: Piton de la Fournaise is not enough?; 53 Marderello torrent: the reference for this net-

[Figure]

work is Coviello et al. (2015); 69 La Colima volcano: please use the international name (Colima Volcano) or the Mexican one (Volcán de Colima); 70 Merapi volcano flanks: please use Merapi volcano, be consistent with the list format; in addition, a number of sites are missing, especially overseas in USA (e.g., Kean et al., 2015), New Zeland (e.g., Lube et al., 2012), and South America (e.g., Kumagai et al., 2009; Worni et al., 2012).

Figure 2: what about adding a sketch of the signal associated to each process?

Figure 13: I guess this is the most important figure of the paper, why does it only appear in the discussion? Given the large seismic dataset I suppose you have at your disposal, why did you plot only between 2 (most of the cases) and 6 (few cases) examples? I wonder if the variability of the attribute shapes is representative given limited number of examples here presented.

References

Coviello, V., Arattano, M., and Turconi, L. (2015). Detecting torrential processes from a distance with a seismic monitoring network. Natural Hazards, 78(3), 2055–2080. https://doi.org/10.1007/s11069-015-1819-2

Gimbert, F., Tsai, V. C., and Lamb, M. P. (2014). A physical model for seismic noise generation by turbulent flow in rivers. Journal of Geophysical Research: Earth Surface, 119, 2209–2238. https://doi.org/10.1002/2014JF003201

Kean, J., Coe, J., Coviello, V., Smith, J., Mccoy, S. W., and Arattano, M. (2015). Estimating rates of debris flow entrainment from ground vibrations. Geophysical Research Letters, 42(15), 6365–6372. https://doi.org/10.1002/2015GL064811

Kumagai, H., Palacios, P., Maeda, T., Castillo, D. B., and Nakano, M. (2009). Seismic tracking of lahars using tremor signals. Journal of Volcanology and Geothermal Research, 183(1–2), 112–121. https://doi.org/10.1016/j.jvolgeores.2009.03.010

Lube, G., Cronin, S. J., Manville, V., Procter, J. N., Cole, S. E., and Freundt,

A. (2012). Energy growth in laharic mass flows. Geology, 40(5), 475–478. https://doi.org/10.1130/G32818.1

Manconi, A., and Coviello, V. (2018). Evaluation of the Raspberry Shakes seismometers to monitor rock fall activity in alpine environments. Geophysical Research Abstracts, Vol. 20, EGU2018-16183.

Occhiena, C., Coviello, V., Arattano, M., Chiarle, M., Morra di Cella, U., Pirulli, M., Pogliotti, P., and Scavia, C. (2012). Analysis of microseismic signals and temperature recordings for rock slope stability investigations in high mountain areas. Natural Hazards and Earth System Science, 12(7), 2283–2298. https://doi.org/10.5194/nhess-12-2283-2012

Schmandt, B., Aster, R. C., Scherler, D., Tsai, V. C., and Karlstrom, K. (2013). Multiple fluvial processes detected by riverside seismic and infrasound monitoring of a controlled flood in the Grand Canyon. Geophysical Research Letters, 40(18), 4858–4863. https://doi.org/10.1002/grl.50953

Schöpa, A., Chao, W.-A., Lipovsky, B., Hovius, N., White, R. S., Green, R. G., and Turowski, J. M. (2017). Dynamics of the Askja caldera July 2014 landslide, Iceland, from seismic signal analysis: precursor, motion and aftermath. Earth Surf. Dynam. Discuss., in review. https://doi.org/10.5194/esurf-2017-68

Worni, R., Huggel, C., Stoffel, M., and Pulgarín, B. (2012). Challenges of modeling current very large lahars at Nevado del Huila Volcano, Colombia. Bulletin of Volcanology, 74(2), 309–324. https://doi.org/10.1007/s00445-011-0522-8

---

## Referee Comment (RC2) · Anonymous Referee #2 · 29 Apr 2018

After reviewing the manuscript I read the review of Dr. Coviello. I fully agree with his comments. I will not repeat his comments in my review.

I found in the manuscript some mistakes and problems. Accordingly, I propose a major revision of the paper (if not rejected in its present form), which is in line with my comments and those of V. Coviello. The document is verbose, with a lot of information (perhaps with little consistency in terms of content) and poor in conclusions. Please, be more concise and remove the unnecessary information. Justify the purpose of the paper better. References must be selected to shorten their number.

In general, I am very skeptical about the purpose of the paper: to establish a standard typology of endogenous seismic sources. It is true that there has been a dramatic increase of monitoring/detecting seismic signals generated by different ground phenom-

ena in the last five years. However, we have to bear in mind that seismic measurements are not a direct measure as they could be extensometers, for example. The terrain is so complex that I am skeptical about whether seismic monitoring could give detailed information about the phenomena. So, seismic data alone are very difficult to manage for mass movement studies, mainly if the signals are very short and are related to small energy release. We must be aware of the type of phenomena. In my opinion, it is the combination of different measurements that can contribute to information about the phenomena. For this paper, I suggest you only include the very significant seismic signals and avoid small events. Mainly, because of the difficulty of a subsequent interpretation. This is in some ways one of the conclusions of the authors, given that they unify the "new named" slopequakes by including them all in one group. The slopequakes can be so complicated that the present catalogue is probably not complete. All efforts in this line should be devoted to monitoring one site with different instruments and to interpreting the events in coordination with different specialists. Having said this, see below for further comments.

1) Table 1. This is a very risky approach. Given the present increase in mass monitored studies you probably miss one unless it is the intention of the authors only to mention those in which they are involved. In this case, you must mention this specifically, and give your reasons.

2) As regards field instrumentation, it would be useful to better explain the characteristics of the instruments and their different site conditions. Site effects are completely ignored in the interpretation/description of the signals. In fact, most of the presented data were already the subject of different interpretations and I assume that these have been described in the corresponding papers. However, when seeking to stablish a standard typology, consideration of the peculiarities (or not) of the site effects is very important.

3) In the definition of the parameters of processing methodology, if I am not mistaken, no amplitudes are considered (only once on pag. 15 line 21).Why are the amplitudes

(nm/s) not indicated in the events? It is true that attenuation can also affect amplitudes, not only the frequency content, but it could be useful for differentiating events. The relative "energy" released together with the duration of the signals can give significance to some events.

4) Additionally, the authors devote a large description to the frequency content of the events. However, in figure 13, the maximum attention is devoted to other parameters that are basically in the time domain. Moreover, the parameters introduced in the processing methodology section are not sufficiently considered in the description of the events. Note that in the description, few of these parameters are mentioned. Furthermore, the last sentence of section 4 merits a detailed explanation and challenges the classification. In this sentence, the authors mention the real problem of the dependence of the defined parameters on the source to sensor distances and on the propagation media properties.

5) Pag. 11. Explanations and description of the signals are very poor. Some explanations correspond to other cases.

I include my comments about the case of RF (pag. 11) only as an example.

Pag. 11 Line 2. Please, indicate if the rock fall was monitored. Information specific on this events is necessary.

Line 5. What does it means. . ... energy below 10 Hz is present for. . ..1 m3 ( Fig 3a). Is this your case, because you mention this figure here?

Line 7. The study of Farin et al., (2014) is an experiment in lab. and cannot be extrapolated to nature as it is observed (the high freq. disappear). This contribution is no relevant here.

Line 9. P- and S- waves are hardly distinguishable. . ... Is this in this specific case? You cannot generalize.

Line 11. First arrivals are mainly impulsive. At the scale of representation I have to

believe it.

Line 12. Figure 4 is incomplete. Information is required. If the signal belongs to a publication, the references must be included. Otherwise a comment is necessary.

Line 13. Why do you suspect that the signals could be different if what you are recording is the movement of the mass falling down the slope? Normally, there is a time lag between ground and blast signals and the signals of the rock fall as observed in earlier publications.

Lines. 18 and 20. Le Roy et al. 2017and 2018. Complete appropriately these references

Line 22. Burtin et al, 2016. This paper is devoted to torrential process. By the way, the reasoning in the outlook section is of interest.

Line 24. You mention "... may be emergent due to simultaneous arrivals of the waves". Explain this better. Do you mean that it could be interference between the impulses? What happens with the wave field? It also depends of the frequency content.

6) Figure 13 is perhaps one of most interesting figures but it must be better explained. As I mentioned before, small events must be avoided.

7) Pag. 16 Line 3-4. The authors justify the differences in the frequency content mentioning attenuation because of large distances, but this is not the case here because it is indicated in the paper and in the abstract that the signals are from events at r<1 km. Is this consistent?

8) The term seismologically is not used correctly in the text. Replace it by seismic instruments. What does seismologically instrumented mean? In the world of seismologists the instruments are seismic instruments or not. They could have different resolution, characteristics, etc. . . "Seismologically" refers to a discipline, but not to the installation. Basically, the parameters you are considering are devoted to data processing signals and signal characteristics and not to wave transmission which is the

subject of seismology. And as regards the installation, what does a non-seismologically installed seismic instrument mean?

9) Pag. 16 line 19 and below. All this information devoted to harmonic signals in the discussion section is out of place. Moreover, it does not correspond to the data presented by you.

10) Discussion. From line 19 to the end. The information provided does not correspond to a discussion of what is presented in the paper. It mainly concerns previous results without comparing them with the data presented in the manuscript. Most of the sentences in the discussion could be included in the introduction, because the information is previous to the results presented in the paper and with little relation to them, at least in the present form. Moreover, I do not understand why the harmonic signals are included in the discussion.

11) As the authors mention on pag. 4 citing (Walter et al., 2017) in MS processing chains (by the way, I do not understand why you include this information in this section): "Many studies approximate the media attenuation field and/or the ground velocity, or do not take into account the topography, leading to mis-location of the events that prevents for accurate interpretation of certain sources and leads to false alarms". Is this the case of the data presented here?

12) Papers under revision although they are public must not be cited, nor must papers in volumes without a standard scientific recognition.

Some comments on the analyses of data.

13) As regards the tremor-like slope-quake (you do not mention this in this way in the title of figure caption of Fig. 12), the PSD is in the range of 8-13Hz, (not 10 Hz as mentioned) and the mean frequency of 20Hz is not clearly deduced from the plots.

14) Slope-quake with harmonic coda (H-SQ). I do not only observe the coda in the Chamousset signal (fig.10a) (note there is an error in this figure) of 08 August, but

also in that of 6 October (fig. 11c). Super-Sauze site slope-quake signal of 24 O ct. (Fig. 10b) and the rock fall signal of 5 Nov (Fig. 4d) also present this behavior. This harmonic coda is present in different events. I think this is significant, and perhaps this is not related to the source but to the site effect for specific frequencies.

15) As regards all figures, but specifically Fig. 3 since it is the reference. Please, indicate the information contained in the plots in the figure caption. What is Amax? Is the parameter defined in section 4? It could be informative to show the maximum amplitude in ground speed units. What are the different traces in different colors shown in plot a? Indicate correctly the power of 10 (10 ˆx and not eˆx).

---

## Author Comment (AC1) · 14 Jun 2018

Dear Editor,

We thank warmly Dr. Velio Coviello and an anonymous referee for their in depth lecture and their many thoughtful and constructive comments. We propose below detailed answers, thoughts and clarification concerning the main points of interrogations of both referees. For clarity, redundant comments of both reviewers and technical/typos comments have been removed or just indicated as OK in the letter.

Sincerely,
Floriane Provost on behalf of all co-authors,

[Figure]

NOTE: In the following document, the referee comments are in normal fonts and the answers are in blue font.

\*\*

**Reviewer 1: Dr. Velio Coviello**

*Major comments:*

I think that the abstract needs a significant rewording. The first lines sound like an introduction on environmental seismology. Please focus more on objectives, methods and results of your work.
Thanks. We have rewritten part of the abstract in order to state more the focus of the work.

In addition, I disagree with the statement "The seismic networks installed on these sites are roughly similar (i.e. sensor, network geometry)". What does "roughly similar" mean? There is a significant difference between a BB seismic network installed on a large, slow moving earth-flow and a linear array of geophones deployed along a debris flow channel.
We do not completely agree with this affirmation. We rephrase the abstract in order to precise that we analyze the signal recorded by geophones and BB seismic sensors in the same frequency band (between ca. 1 Hz and 100 Hz). We do not investigate the information recorded at lower (BB) or higher (Geophones) frequencies. In order to ease the understanding, we also propose a new table (Table 2) presenting the list and the specifications of the seismic instruments and of the seismic network geometry of

the 13 sites where data is presented and analyzed.

Moreover, if the authors are focusing on "seismic events detected at close distances (< 1 km)" the sensor network characteristics and geometry, as well as the geological and geomorphological contexts, have a strong impact on the recorded signals.

Indeed, but this statement is true for every seismological studies. More than the distance alone it is the wavelength of the seismic waves and the source dimension compared to the recording distance that is important. We analyzed seismic networks where at least one sensor is installed on or at a very close vicinity (< 50 m) to the active zone. Regarding the geological and geomorphological contexts, our assumption is that if we can observe similar signal features in different sites they can only be explained by the similarity of seismic sources.

To achieve a standard source characterization, in my opinion there are three major topics that would need to be addressed: i) distance sensor-source, ii) typology of sensor, iii) sensor installation methods. Given the pretty ambitious title, I would expect some discussion of their effects on landslide sources.

As mentioned in the previous comments, we analyzed seismic networks where at least one sensor is installed on or at a very close vicinity (< 50 m) to the active zone and we also filter the signals in the same (low) frequency band to limit the influence of the wave propagation of the signal.

Concerning point ii), to compare signals from different networks the most important sensor-related properties to take into account is the instrumental response of the sensors. For each case presented in our study, we have removed the instrumental response of the recorded signals (and filtered the signal in the same frequency band ($f_c$ to 50 Hz) recorded by every sensors in our dataset to compute quantitatively their properties).

Concerning point iii), if the reviewer means network geometry by "sensor installation"

we do not agree that the sensor installation will play an important role in the signal features. The latter is mostly controlled by the source to sensor distance (and we answer to this comment in point i)). The sensor installation will play an important role in the magnitude of completeness and in the location accuracy.

We have added in the Table 2 some information on the distance sensor-sources for each case studies in order to provide more information about the analyzed datasets. We state clearly that we do not investigate low and high frequencies (P10, lines 12 to 15). As we choose highly energetic examples for each class we do not expect a dominant impact of site effect on the features we selected and we discussed if needed be, the effect on the interpretation.

However, I have the impression that the paper leaves more open questions than clear responses. In the following more details on how these three aspects have not been adequately addressed are given. i) The authors briefly touch this point in the discussion: "The differences in the frequency content of simple slopequakes may be explained either by the attenuation of the high frequency at large distances during the propagation or by different rupture velocity and/or the presence of fluid in the fault plane". I encourage them to stress more on the possible limitations of a spectral analysis to be employed in a general classification. For instance, consider what was already published about the effect of the sensor-source distance on the seismic signal produced by flow processes (Gimbert et al., 2014; Schmandt et al., 2013).

We agree that spectral analysis of the seismic signals present some limitations for signal comparison but it is also the most common approach to investigate seismic datasets. Spectral analysis is used in most classification processes (automated and manual) whether it is for volcano or reservoir monitoring, local, regional or even global seismology.

It must be noted that 1) we do not only analyze the spectrum (4 over 9 of the signals properties are not directly correlated to the spectral content), 2) in order to reduce the influence of the seismic to sensor distance, the signals are filtered in the same

frequency band ($<$ 50 Hz) before the computation of the features (this is not the case on the signal figures) and 3) we analyze the most energetic recorded signals in order to reduce the influence of the seismic geometry.

Concerning the two mentioned studies, they show that the source mechanism is a predominant factor controlling signal spectral content although the sensor to source distance plays a role in the contribution of certain frequencies to the signals amplitude. As the simplest deconvolutive model, propagation acts as a filter, but the remaining spectral content is controlled by source properties. Hence, even if we loose spectral information due to attenuation, the peculiarity of the spectrum controlled by the source mechanism is most of the time conserved. Therefore we think that including spectral features is relevant in our classification.

ii) At P7 L5-6 the authors state "The relatively low energy released by the land-slide related sources makes the choice of the seismological instruments to deploy very important". I agree, and I think that this point should be developed more.

We added a section about the seismic network deployment where we address this comment. We also modify the last paragraph of section 3.1 (P6. l32, P7. l13).

Section 3.1 describes the main classes of sensors employed for the detection of mass movements but I do not see a proper discussion of this point when the authors present their dataset.

Thanks. We refer to the new section "Data" introducing Table 2, and we also indicate more explicitly which sensor types are used and how they are analyzed. As mentioned previously, we corrected every sensor response and we decided to work in the $f_c$-50 to 100 Hz frequency band were all analyzed sensors are sensitive.

iii) Considering flow detection at channel scale, the sensor installation method has a strong impact on the features of the recorded signal, both in amplitude (e.g., Coviello et al., 2015) and frequency domain (e.g., supplementary material of Kean

et al., 2015). Again, this issue is shortly introduced at P3 L11 "The location of the sensors and the type of waveguide are also critical to capture the slope behavior" but a discussion based on the analyzed dataset is missing.

We added a section about the seismic network deployment (Section 3.2) where we address this comment. More than the sensor installation geometry is the distance to the source that plays an important role in the recorded amplitude and the frequency content. We already answer about this influence in previous comments.

Standardized datasets and field experiments are probably needed to systematically address those topics. I am skeptical about the possibility to develop a standardized source-mechanisms characterization of landslide-induced seismic signals from a collection of heterogeneous case studies.

We are a bit confused by this comment. On one hand the reviewer stresses that "standardized dataset are probably needed" but on the other hand that it is impossible to do so from a collection of case studies. Then how can one compile standardized datasets?

We believe that the compilation of case studies and the standardized processing and representation of the seismic events recorded on landslides we propose is relevant for the following reasons : 1) standardized classifications exist in other fields of micro-seismology such as in reservoir monitoring, slow earthquakes (LFE, VLFE, etc) and volcano monitoring; 2) Those standardized classifications have proven to be useful starting points for further discussions : the classification is never frozen and should evolve following new observations and models; 3) Compiling datasets from very diverse case studies allow to bring out the control of the source on the signals from each class (different media and different propagation paths but same signal characteristics at different sites = source-controlled features).

*Additional comments:*

P2 L5: references needed
OK. We have introduced different references for glaciers, snow avalanches and landslides.

P2 from L26: concerning repeaters, I would suggest to the authors to read the reviews of the paper by Schopa et al. (2017), an interesting discussion is made there on this point
We added a sentence concerning this discussion (P3. l15-20).

P3 L16: "low frequency ranges (1-500 Hz)", why do you define this pretty broad frequency range as low? Compared to what?
We recognize that this sentence is awkward. We meant compare to Acoustic Emission signals. The term "low frequency" has been removed for clarity.

P4 L30: "13 monitored sites", 13 or 14?
OK. The correct number of sites is 13.

P4 L33-35: concerning "we first discuss all the physical processes that occur on landslides: We further present the seismologically-instrumented landslides in the world: Then we establish a classification scheme", I suggest the authors to rephrase in order to be more realistic. I think that the main physical processes were discussed, that only a few (14) of the seismologically-instrumented landslides in the world were presented and that a possible classification scheme was proposed.
We have rephrased the sentence: *"Then we establish a classification scheme of the landslide seismic signals from relevant signal features based on the analysis of the datasets of 13 sites."*

P8 L13-24: these lines sound more as part of introduction than data. The paragraph data should start from current L25 but a description of the analyzed dataset is

actually missing: which is the length of the analyzed time series? How many events did you analyze? How did you select the events analyzed in the following paragraphs? I guess those were well-known events, or did you applied an automatic detection methods?

We added these information in the new version of the paper and in Table 2, section 4 and section 5 of the new version.

P8 L26: "For all sites, the instruments are deployed close to the landslide", what does "close" mean? Please be more specific. I guess that authors agree that, for example, two seismic sensors, one installed at 10 m and another one at 900 m from the very same landslide, would record signals pretty different, especially in terms of spectral content.

We added these informations in the description of the sites and in Table 2 and section 4. We mentioned that for each seismic network analyzed, at least one sensor is installed on the active zone or at its vicinity ($< 50$ m). Moreover, we choose to work with the most energetic trace for each recorded events that we assume to be the closest station to the source and hence, the most representative of the seismic sources properties.

Of course, the distance and the medium contributes in the features of the seismic signals and we do not decorrelate its contribution. But as mentioned in earlier answer, the source mechanism also contribute to the signals feature. We already justified our approach to limit the wave propagation influence (see earlier answers to the same comment). Basically, our assumption is (as mentioned in previous comment): different media and different propagation paths but same characteristics at different sites = source-controlled features.

P15 L14: "The signals present significant differences with the chosen features", please reword, the reader does not understand the meaning of this sentence.

We have rephrased the sentence.

P15 L15: "in the field, the differentiation", I am not sure to have correctly understood, maybe you meant "only from the seismic signal analysis, the difference between"?
OK. We have rephrased the sentence.

P17 L23: please avoid references that are not published work, i.e. Helmstetter et al., (2017a), especially if the reference is used to support a very strong statement such as "the high correlation between the repetitive events could only be explained by stick-slip movement of the locked section(s)". A sentence like this must be accompanied by supporting data or published results.
We removed the mentioned reference in this sentence.

P17 L29: concerning "most collapses occurred without precursory sequences (Allstadt et al., 2017)", I would suggest tuning down this statement, which is also in contradiction with P2 L24. There are a number of cases where precursory seismic signals related to small rockfalls were documented, especially when a station is installed nearby the slope or there is a local monitoring network. On the contrary, when the closest station is distant or we do not dispose of other monitoring data, recognizing those precursory events is difficult but potentially there are. I also believe that the reference Allstadt et al. (2017) is not consistent here.
OK. We agree and removed this sentence.

P18 L16-20: I do agree with "several descriptions of the seismic sources are proposed for each study case" and a standard classification would help to discuss and compare landslide-induced seismic signals. I understand that the authors are proposing their classification as general reference, but I would suggest to the authors to delete the sentence "we encourage future studies to use and possibly enrich the proposed typology". In my opinion the scientific community does not need to be

encouraged to adopt one classification or another.

We disagree with this statement – standard typology does exist for instance for volcano-related seismic sources or for glacier-related sources and have been very useful to progress in the comparison of the seismic signals on all volcanoes and in the creation of comparable catalogs. Though any standardization/harmonization methods can be questionable, we believe that proposing a nomenclature of sources is important for further discussions including rejecting the proposed classification or interpretation.

By the way, why you do not adopt the classification proposed by Allstadt et al. (2017)?

The classification proposed by Allstadt is not comparable to our classification as it is related to detection and cataloging landslide failures at regional scales (> 1 km); the purpose of our classification is the slope scale.

P18 L25-27: reference needed
Done. We added the reference.

Reference list: the style is not consistent with the journal guidelines, in many cases the doi is missing, there are repeated references (Hibert et al., 2014a), others are missing (Provost et al., 2018) and there is some text here and there probably out of place (e.g., P29 L10-11). An accurate revision of the reference list is needed.

We corrected the style of the references taking into account the journal guidelines, and also updated the reference list.

Moreover, I do acknowledge the significant contribution of some of the authors to the field but I have the impression that self-citations are really abundant (five papers by Hibert et al., six by Helmstetter et al.). Please try to select your most significant works and refer to them.

We believe the citations related to the papers of Helmstetter and Hibert are relevant.

We also added several new references to the manuscript from other research groups as proposed by reviewer # 1. We tried to be exhaustive in the references and we cite more than 130 papers (a significant number due to Table 1), in total around 18% of the citations are self-citations of the co-authors which we think is not over-abundant.

Figure 1: I would prefer the author to focus more on the sites from which they present some data instead of showing a collection of points in a global map. In addition, Figure 1 is redundant if one considers the list presented in Table 1.
We added a table gathering the informations about the analyzed sites and their seismic networks (Table 2). We removed Figure 1.

Table 1: some details/revisions are needed. 7 Alestch-Moosfluh: this landslide is also monitored with a geophone network (Manconi and Coviello, 2018); 8 Torgiovannetto, Assise: please modify in Assisi; 15 Aiguilles: Aiguilles Pas de l'Ours?; 22 US highway 50, CA: there is no reference/website about that?; 24 Millcoma Meander, Oregon: same as above; 33 Matterhorn peak/Mont Cervin: please use the international name (Matterhorn) or the Italian one (Cervino) and add the reference describing the more recent monitoring network (Occhiena et al., 2012); 48 Piton de la Fournaise caldeira: Piton de la Fournaise is not enough?; 53 Marderello torrent: the reference for this net- work is Coviello et al. (2015); 69 La Colima volcano: please use the international name (Colima Volcano) or the Mexican one (Volcán de Colima); 70 Merapi volcano flanks: please use Merapi volcano, be consistent with the list format; in addition, a number of sites are missing, especially overseas in USA (e.g., Kean et al., 2015), New Zeland (e.g., Lube et al., 2012), and South America (e.g., Kumagai et al., 2009; Worni et al., 2012).
OK. Thanks for providing this detailed information. We have corrected the Figure and Table 1 accordingly.

Figure 2: what about adding a sketch of the signal associated to each process?

We do not think this would had information at this stage of the paper. It seems to us that simple sketching cannot capture the complexity of seismic signals and that the representation we propose on figure 13 is more suited to expose this complexity.

Figure 13: I guess this is the most important figure of the paper, why does it only appear in the discussion?
This figure summarizes the presented signals properties. We do not think that an earlier presentation of this figure is necessary.

Given the large seismic dataset I suppose you have at your disposal, why did you plot only between 2 (most of the cases) and 6 (few cases) examples? I wonder if the variability of the attribute shapes is representative given limited number of examples here presented.
We present more examples in the discussion in the new version of the paper (Figure 12) and comment the variability of the attributes in the discussion (P17. l2-15).

**

**Reviewer 2: anonymous referee**

After reviewing the manuscript I read the review of Dr. Coviello. I fully agree

with his comments. I will not repeat his comments in my review. I found in the manuscript some mistakes and problems. Accordingly, I propose a major revision of the paper (if not rejected in its present form), which is in line with my comments and those of V. Coviello. The document is verbose, with a lot of information (perhaps with little consistency in terms of content) and poor in conclusions. Please, be more concise and remove the unnecessary information. Justify the purpose of the paper better. References must be selected to shorten their number.

We thank the reviewer for these statements. The introduction of the paper has been thoroughly revised to better highlight the focus of the work. We also rephrased or deleted some sentences considered as verbose by the reviewer. All these changes are indicated in track mode changes in the revised version of the manuscript.

In general, I am very skeptical about the purpose of the paper: to establish a standard typology of endogenous seismic sources.

This comment has also been addressed by Reviewer #1. We believe that the compilation of case studies and the standardized processing and representation of the seismic events recorded on landslides we propose is relevant for the following reasons : 1) standardized classifications exist in other fields of micro-seismology such as in reservoir monitoring, slow earthquakes (LFE, VLFE, etc) and volcano monitoring; 2) Those standardized classifications have proven to be useful starting points for further discussions : the classification is never frozen and should evolve following new observations and models; 3) Compiling datasets from very diverse case studies allow to bring out the control of the source on the signals from each class (different media and different propagation paths but same signal characteristics at different sites = source-controlled features).

It is true that there has been a dramatic increase of monitoring/detecting seismic signals generated by different ground phenomena in the last five years. However, we have to bear in mind that seismic measurements are not a direct measure as

they could be extensometers, for example. The terrain is so complex that I am skeptical about whether seismic monitoring could give detailed information about the phenomena.

We disagree with this statement. The arguments for using seismology, waveform analysis and analysis of the temporal and spatial distribution of seismic sources on landslides as complementary sources of information on the mechanics of the processes are:

1) The temporal resolution of seismic instruments provides very accurate timing of the deformation processes and is non-invasive as it can detect events at distance from the sensor installation. These advantages are hardly met simultaneously with other types of sensor. Several studies have demonstrated the major contribution of seismology to built near-exhaustive catalogs of events at slope scale (Helmstetter et al, 2011; Dietze et al, 2017b), at regional scale (Hammer et al, 2016) and its potential for early-warning of debris-flows (Walter et al, 2017; Arratano et al., 1999; Burtin et al., 2009).

2) It records also the spatial distribution of the sources occurring in depth (Spillmann et al, 2007, Lacroix et al, 2011, Tang et al, 2015) which is not the case of extensometers for example. The location of the seismic activity represent valuable information to update geo-mechanical models determining the factor of safety of the slope (Tang et al., 2015).

3) The seismic signal features are controlled by the source mechanism providing insights in the mechanical behavior of the deformation.

4) Recent papers have also documented seismic signatures preceding the collapse of large landslides (Amittrano et al., 2005; Yamada et al., 2016; Poli 2017; Schöpa et al., 2018) proving the presence of seismic signals associated to slope instabilities deformation.

So, seismic data alone are very difficult to manage for mass movement studies, mainly if the signals are very short and are related to small energy release. We must be aware of the type of phenomena. In my opinion, it is the combination of different

measurements that can contribute to information about the phenomena.
We agree and we never mentioned to consider seismology as a standalone technique
for landslide monitoring.

For this paper, I suggest you only include the very significant seismic signals
and avoid small events.
The purpose of the paper is clearly mentioned: we analyze the signals recorded at
close distance to the slope ($<$ 1km) with seismic sensor sensitive to the $\tilde{1}$-100 Hz fre-
quency band. This means that we are exploring larger events and more distant events
than Acoustic Emission studies (Dixon et al, 2015; Michlmayr, 2012) but smaller events
than large slope failure (volume $>$ 1O$^6$ m$^3$, Ekstrom and Stark, 2013). Obviously, the
examples are "significant" signals at this scale as they are clearly above the noise level.

Mainly, because of the difficulty of a subsequent interpretation. This is in some
ways one of the conclusions of the authors, given that they unify the "new named"
slopequakes by including them all in one group. The slopequakes can be so compli-
cated that the present catalogue is probably not complete.
We agree with the statement that slopequake signals can be "complicated" and
"that the present catalogue is probably not complete". However, we also propose
sub-classes taking into account the complexity of this class while keeping a uniform
denomination because they are usually analyzed as one class in the previous and
current studies. The name "slopequake" was chosen in order to remove the source
mechanisms interpretation induced by the name "slidequake" or "micro-earthquake".
As mentioned earlier, the present classification is not frozen and can be enriched
and/or discussed. In particular, for certain sub-classes we explicitly mentioned that
surface processes may also generate these type of signals (SQ-tremor like signals
and SQ-with precursory).

All efforts in this line should be devoted to monitoring one site with different instruments and to interpreting the events in coordination with different specialists.
This is done on most of the recent sites being instrumented by different research groups. However, the source mechanisms and the variability of the slopequake features remain poorly documented. Understanding this variability and the underlying physical processes remains a strong challenge, and we hope that the classification we propose will bring some insights leading toward a better grasp of those processes.

*Having said this, see below for further comments.*

1) Table 1. This is a very risky approach. Given the present increase in mass monitored studies you probably miss one unless it is the intention of the authors only to mention those in which they are involved. In this case, you must mention this specifically, and give your reasons.
We agree that we probably missed some sites especially the new sites recently instrumented. We added the missed sites suggested by V. Coviello (4 over 70). If we are missing further references, please, let us know and we will add them to the table.

2) As regards field instrumentation, it would be useful to better explain the characteristics of the instruments and their different site conditions. Site effects are completely ignored in the interpretation/description of the signals. In fact, most of the presented data were already the subject of different interpretations and I assume that these have been described in the corresponding papers. However, when seeking to establish a standard typology, consideration of the peculiarities (or not) of the site effects is very important.
This comment was already addressed by Reviewer #1.
Concerning the field instrumentation, we propose a new section "Data" (section 4) to describe precisely the seismic network configuration (also summarized in table 2). The geomorphological and geological context are indicated in Table 1 with the references for further information.

Concerning the instruments, we corrected the instrument response and analyzed their common frequency band.

Concerning the site effects, it is true that we do not correct. However, we believe that the comparison of signals from different sites of various geomorphological and geological contexts is precisely a good strategy to discriminate the contribution of the source mechanism from the site effects/attenuation. We hence describe the features shared by all the selected examples without focusing on particular features of certain signals that are likely linked to site effects.

3) In the definition of the parameters of processing methodology, if I am not mistaken, no amplitudes are considered (only once on pag. 15 line 21). Why are the amplitudes (nm/s) not indicated in the events? It is true that attenuation can also affect amplitudes, not only the frequency content, but it could be useful for differentiating events. The relative "energy" released together with the duration of the signals can give significance to some events.

Amplitudes are indicated on the figures for each trace of each example in nm/s. We did not choose to analyze the amplitudes or Energy/duration relationships (even so, they can be significantly different from one class to the other) because we are focusing in the features that can be related to the source mechanisms and not to its magnitude.

4) Additionally, the authors devote a large description to the frequency content of the events. However, in figure 13, the maximum attention is devoted to other parameters that are basically in the time domain.

Over 9 parameters presented on figure 13, 5 are related to the frequency content and 4 to the time domain. The waveforms presented on figure 13 gather information on both time and frequency content. Moreover, we think that the format of the figure used to present examples for each classes (Figures 3 to 12) summarizes all the informations needed to discuss the signals. On Figure 13, we adopted "star" diagrams in order to ease the visualization of all the selected features and not to focus only on frequency

nor time domain properties.

Moreover, the parameters introduced in the processing methodology section are not sufficiently considered in the description of the events. Note that in the description, few of these parameters are mentioned.
OK. We reviewed these sections to add these informations.

Furthermore, the last sentence of section 4 merits a detailed explanation and challenges the classification. In this sentence, the authors mention the real problem of the dependence of the defined parameters on the source to sensor distances and on the propagation media properties.
OK. We have rewritten this part of section 4 (Section 5. in the revised manuscript) to describe our approach to analyze the datasets and compare them (P10. l21-30).

5) Pag. 11. Explanations and description of the signals are very poor. Some explanations correspond to other cases. I include my comments about the case of RF (pag. 11) only as an example.
We reviewed these sections to add further informations.

Pag. 11 Line 2. Please, indicate if the rockfall was monitored. Information specific on this event is necessary.
OK. We added information about this event on the description of the datasets.

Line 5. What does it means:. energy below 10 Hz is present for volume larger than 1 m$^3$ (Fig 3a). Is this your case, because you mention this figure here?
It is not only the case of this specific event. We added references to support this statement in the next paragraph: *"The frequency content is also controlled by the block mass i.e. the frequency of spectral maximum energy decreases when the block mass increases (Farin et al, 2015; Burtin et al, 2016; Huang et al, 2007".*

Line 7. The study of Farin et al., (2014) is an experiment in lab. and cannot be extrapolated to nature as it is observed (the high freq. disappear). This contribution is no relevant here.
OK. We removed this statement.

Line 9. P- and S- waves are hardly distinguishable. Is this in this specific case? You cannot generalize.
The statement is supported by different references.

Line 11. First arrivals are mainly impulsive. At the scale of representation I have to believe it.
We removed statements concerning the impulsive nature of the signals as it may vary from site to site.

Line 12. Figure 4 is incomplete. Information is required. If the signal belongs to a publication, the references must be included. Otherwise a comment is necessary.
The reference of the dataset from which the presented events are taken from are added in the caption when the presented (or similar events) have been published.

Line 13. Why do you suspect that the signals could be different if what you are recording is the movement of the mass falling down the slope? Normally, there is a time lag between ground and blast signals and the signals of the rock fall as observed in earlier publications.
We meant that natural rockfall are often composed of several falling blocks subject to break-up. In the case of the Riou-Bourdoux experiment only single block falls were monitored. It is true that in other studies when the rockfall is triggered from the rock cliff, very similar mechanisms and signals can be observed. We hence rephrased the sentence accordingly.

Lines. 18 and 20. Le Roy et al. 2017and 2018. Complete appropriately these references

OK. We have corrected the references.

Line 22. Burtin et al, 2016. This paper is devoted to torrential process. By the way, the reasoning in the outlook section is of interest.

Fig1. of Burtin et al, 2016 shows the influence of the block mass on the frequency content even so, it is not discussed in the text. We added Huang et al, 2007 that discusses the same experiment.

Line 24. You mention "[...] may be emergent due to simultaneous arrivals of the waves". Explain this better. Do you mean that it could be interference between the impulses? What happens with the wave field? It also depends of the frequency content.

OK. We rephrased the sentence as: " [...] may be emergent due to simultaneous arrivals of waves generated by impactors of different sizes impacting the ground at closely spaced time intervals".

6) Figure 13 is perhaps one of most interesting figures but it must be better explained. As I mentioned before, small events must be avoided.

We enriched the discussion of this figure (P.16 l34 to P.17 l.7). We already respond to the "small events" issue in a previous comment. Basically, we selected events clearly above the noise level.

7) Pag. 16 Line 3-4. The authors justify the differences in the frequency content mentioning attenuation because of large distances, but this is not the case here because it is indicated in the paper and in the abstract that the signals are from events at r< 1 km. Is this consistent?

Attenuation is function of the distance and the wavelength of the seismic waves observed. c.f. previous response to comments of Reviewer #1. At our scale, "large distances" ranges from 100 m and more, depending on the magnitude of the source and the network geometry. One can clearly observed the influence of the distance in most the presented examples (Figures 3 to 11), even if the location of the source is not computed. Moreover, this comment is in contradiction with all the previous comments concerning the influence of the wave propagation on the recorded signals as a strong limitation of our study.

8) The term seismologically is not used correctly in the text. Replace it by seismic instruments. What does seismologically instrumented mean? In the world of seismologists the instruments are seismic instruments or not. They could have different resolution, characteristics, etc: : : "Seismologically" refers to a discipline, but not to the installation. Basically, the parameters you are considering are devoted to data processing signals and signal characteristics and not to wave transmission which is the subject of seismology. And as regards the installation, what does a non-seismologically installed seismic instrument mean?
OK. We corrected accordingly.

9) Pag. 16 line 19 and below. All this information devoted to harmonic signals in the discussion section is out of place. Moreover, it does not correspond to the data presented by you.
We present data from our gathered dataset. Except the one recorded at the Slumgullion landslide (Gomberg et al.), none of these signals have been published before. We discuss why we do not refer to these signals in the proposed classification.

10) Discussion. From line 19 to the end. The information provided does not correspond to a discussion of what is presented in the paper. It mainly concerns previous results without comparing them with the data presented in the manuscript.

We have thoroughly rewritten the discussion section.

Most of the sentences in the discussion could be included in the introduction, because the information is previous to the results presented in the paper and with little relation to them, at least in the present form.
We have rewritten the introduction and the discussion to take into account this comment and the previous ones.

Moreover, I do not understand why the harmonic signals are included in the discussion.
We discussed the harmonic signals in the Discussion section as we are not including them in the proposed classification whereas they have been presented in other studies. We find surprising that, on the one hand, reviewers reproach us not to compare our data to other and then, on the other hand, find the paragraph where we do this comparison not relevant.

11) As the authors mention on pag. 4 citing (Walter et al., 2017) in MS processing chains (by the way, I do not understand why you include this information in this section): "Many studies approximate the media attenuation field and/or the ground velocity, or do not take into account the topography, leading to mislocation of the events that prevents for accurate interpretation of certain sources and leads to false alarms". Is this the case of the data presented here?
We talked about location of the source which is an important information to associate the recorded signals to slope deformation. However, we mentioned here that location using attenuation law and assuming a homogeneous attenuation factor may lead to mislocation of the seismic events. Consequently, if the location error is of the same order of the distribution of geomorphological structures, it can be difficult to interpret the source of the recorded signals.
In the present study we did not locate the events and focus on the signals features

that can be related to the seismic source mechanism. The later is discussed in each sub-classes presentation with reference to studies that modeled the seismic sources from the seismic signals or to studies that observe similar signals in different context (e.g. glacier motion).

12) Papers under revision although they are public must not be cited, nor must papers in volumes without a standard scientific recognition.
We cite posters and abstracts only to present the monitoring sites and/or the datasets (Table 1 and Figure 2 to 10). We removed reference to posters/abstract when supporting statements in the text. For the papers under revision, we let the editor decides whether they should be included in the reference list (most of them being today accepted).

Some comments on the analyses of data. 13) As regards the tremor-like slope-quake (you do not mention this in this way in the title of figure caption of Fig. 12), the PSD is in the range of 8-13Hz, (not 10 Hz as mentioned) and the mean frequency of 20Hz is not clearly deduced from the plots.
OK. We corrected the description of this class and the caption.

14) Slope-quake with harmonic coda (H-SQ). I do not only observe the coda in the Chamousset signal (fig.10a) (note there is an error in this figure) of 08 August, but also in that of 6 October (fig. 11c). Super-Sauze site slope-quake signal of 24 Oct. (Fig. 10b) and the rock fall signal of 5 Nov (Fig. 4d) also present this behavior. This harmonic coda is present in different events. I think this is significant, and perhaps this is not related to the source but to the site effect for specific frequencies.
We agree that for this particular case, wave propagation could be a better explanation for the signal feature. Consequently, we removed this class from the classification and we discuss this signal feature in the new version of the discussion.

15) As regards all figures, but specifically Fig. 3 since it is the reference. Please, indicate the information contained in the plots in the figure caption. What is Amax? Is the parameter defined in section 4? It could be informative to show the maximum amplitude in ground speed units. What are the different traces in different colors shown in plot a? Indicate correctly the power of 10 (10 ËĘx and not eËĘx).

OK. We indicated that $A_{max}$ refers to the maximum amplitude (nm/s). The different traces in different colors correspond to the other sensors present on the site, we added this comment in the caption. We modify the power accordingly.

**Supplement:**

Dear Editor,

We thank warmly Dr. Velio Coviello and an anonymous referee for their in depth lecture and their many thoughtful and constructive comments. We propose below detailed answers, thoughts and clarification concerning the main points of interrogations of both referees. For clarity, redundant comments of both reviewers and technical/typos comments have been removed or just indicated as OK in the letter.

Sincerely,
Floriane Provost on behalf of all co-authors,

NOTE: In the following document, the referee comments are in normal fonts and the answers are in blue font.

**

**Reviewer 1: Dr. Velio Coviello**

*Major comments:*

I think that the abstract needs a significant rewording. The first lines sound like an introduction on environmental seismology. Please focus more on objectives, methods and results of your work.
Thanks. We have rewritten part of the abstract in order to state more the focus of the work.

In addition, I disagree with the statement "The seismic networks installed on these sites are roughly similar (i.e. sensor, network geometry)". What does "roughly similar" mean? There is a significant difference between a BB seismic network installed on a large, slow moving earth-flow and a linear array of geophones deployed along a debris flow channel.
We do not completely agree with this affirmation. We rephrase the abstract in order to precise that we analyze the signal recorded by geophones and BB seismic sensors in the same frequency band (between ca. 1 Hz and 100 Hz). We do not investigate the information recorded at lower (BB) or higher (Geophones) frequencies. In order to ease the understanding, we also propose a new table (Table 2) presenting the list and the specifications of the seismic instruments and of the seismic network geometry of the 13 sites where data is presented and analyzed.

Moreover, if the authors are focusing on "seismic events detected at close distances (< 1 km)" the sensor network characteristics and geometry, as well as the geological and geomorphological contexts, have a strong impact on the recorded signals.

Indeed, but this statement is true for every seismological studies. More than the distance alone it is the wavelength of the seismic waves and the source dimension compared to the recording distance that is important. We analyzed seismic networks where at least one sensor is installed on or at a very close vicinity ($< 50$ m) to the active zone. Regarding the geological and geomorphological contexts, our assumption is that if we can observe similar signal features in different sites they can only be explained by the similarity of seismic sources.

To achieve a standard source characterization, in my opinion there are three major topics that would need to be addressed: i) distance sensor-source, ii) typology of sensor, iii) sensor installation methods. Given the pretty ambitious title, I would expect some discussion of their effects on landslide sources.

As mentioned in the previous comments, we analyzed seismic networks where at least one sensor is installed on or at a very close vicinity ($< 50$ m) to the active zone and we also filter the signals in the same (low) frequency band to limit the influence of the wave propagation of the signal.

Concerning point ii), to compare signals from different networks the most important sensor-related properties to take into account is the instrumental response of the sensors. For each case presented in our study, we have removed the instrumental response of the recorded signals (and filtered the signal in the same frequency band ($f_c$ to 50 Hz) recorded by every sensors in our dataset to compute quantitatively their properties).

Concerning point iii), if the reviewer means network geometry by "sensor installation" we do not agree that the sensor installation will play an important role in the signal features. The latter is mostly controlled by the source to sensor distance (and we answer to this comment in point i)). The sensor installation will play an important role in the magnitude of completeness and in the location accuracy.

We have added in the Table 2 some information on the distance sensor-sources for each case studies in order to provide more information about the analyzed datasets. We state clearly that we do not investigate low and high frequencies (P10, lines 12 to 15). As we choose highly energetic examples for each class we do not expect a dominant impact of site effect on the features we selected and we discussed if needed be, the effect on the interpretation.

However, I have the impression that the paper leaves more open questions than clear responses. In the following more details on how these three aspects have not been adequately addressed are given. i) The authors briefly touch this point in the discussion: "The differences in the frequency content of simple slopequakes may be explained either by the attenuation of the high frequency at large distances during the propagation or by different rupture velocity and/or the presence of fluid in the fault plane". I encourage them to stress more on the possible limitations of a spectral analysis to be employed in a general classification. For instance, consider what was already published about the effect of the sensor-source distance on the seismic signal produced by flow processes (Gimbert et al., 2014; Schmandt et al., 2013).

We agree that spectral analysis of the seismic signals present some limitations for signal comparison but it is also the most common approach to investigate seismic datasets. Spectral analysis is used in most classification processes (automated and manual) whether it is for volcano or reservoir monitoring, local, regional or even global seismology.

It must be noted that 1) we do not only analyze the spectrum (4 over 9 of the signals properties are not directly correlated to the spectral content), 2) in order to reduce the influence of the seismic to sensor distance, the signals are filtered in the same frequency band ($< 50$ Hz) before the computation of the features (this is not the case on the signal figures) and 3) we analyze the most energetic recorded signals in order to reduce the influence of the seismic geometry.

Concerning the two mentioned studies, they show that the source mechanism is a predominant factor controlling signal spectral content although the sensor to source distance plays a role in the contribution of certain frequencies to the signals amplitude. As the simplest deconvolutive model, propagation acts as a filter, but the remaining spectral content is controlled by source properties. Hence, even if we loose spectral information due to attenuation, the peculiarity of the spectrum controlled by the source mechanism is most of the time conserved. Therefore we think that including spectral features is relevant in our classification.

ii) At P7 L5-6 the authors state "The relatively low energy released by the landslide related sources makes the choice of the seismological instruments to deploy very important". I agree, and I think that this point should be developed more.

We added a section about the seismic network deployment where we address this comment. We also modify the last paragraph of section 3.1 (P6. l32, P7. l13).

Section 3.1 describes the main classes of sensors employed for the detection of mass movements but I do not see a proper discussion of this point when the authors present their dataset.

Thanks. We refer to the new section "Data" introducing Table 2, and we also indicate more explicitly which sensor types are used and how they are analyzed. As mentioned previously, we corrected every sensor response and we decided to work in the $f_c$-50 to 100 Hz frequency band were all analyzed sensors are sensitive.

iii) Considering flow detection at channel scale, the sensor installation method has a strong impact on the features of the recorded signal, both in amplitude (e.g., Coviello et al., 2015) and frequency domain (e.g., supplementary material of Kean et al., 2015). Again, this issue is shortly introduced at P3 L11 "The location of the sensors and the type of waveguide are also critical to capture the slope behavior" but a discussion based on the analyzed dataset is missing.

We added a section about the seismic network deployment (Section 3.2) where we address this comment. More than the sensor installation geometry is the

distance to the source that plays an important role in the recorded amplitude and the frequency content. We already answer about this influence in previous comments.

Standardized datasets and field experiments are probably needed to systematically address those topics. I am skeptical about the possibility to develop a standardized source-mechanisms characterization of landslide-induced seismic signals from a collection of heterogeneous case studies.

We are a bit confused by this comment. On one hand the reviewer stresses that "standardized dataset are probably needed" but on the other hand that it is impossible to do so from a collection of case studies. Then how can one compile standardized datasets?

We believe that the compilation of case studies and the standardized processing and representation of the seismic events recorded on landslides we propose is relevant for the following reasons : 1) standardized classifications exist in other fields of micro-seismology such as in reservoir monitoring, slow earthquakes (LFE, VLFE, etc) and volcano monitoring; 2) Those standardized classifications have proven to be useful starting points for further discussions : the classification is never frozen and should evolve following new observations and models; 3) Compiling datasets from very diverse case studies allow to bring out the control of the source on the signals from each class (different media and different propagation paths but same signal characteristics at different sites = source-controlled features).

*Additional comments:*

P2 L5: references needed

OK. We have introduced different references for glaciers, snow avalanches and landslides.

P2 from L26: concerning repeaters, I would suggest to the authors to read the reviews of the paper by Schopa et al. (2017), an interesting discussion is made there on this point

We added a sentence concerning this discussion (P3. l15-20).

P3 L16: "low frequency ranges (1-500 Hz)", why do you define this pretty broad frequency range as low? Compared to what?

We recognize that this sentence is awkward. We meant compare to Acoustic Emission signals. The term "low frequency" has been removed for clarity.

P4 L30: "13 monitored sites", 13 or 14?

OK. The correct number of sites is 13.

P4 L33-35: concerning "we first discuss all the physical processes that occur on landslides: We further present the seismologically-instrumented landslides in the world: Then we establish a classification scheme", I suggest the authors to

rephrase in order to be more realistic. I think that the main physical processes were discussed, that only a few (14) of the seismologically-instrumented landslides in the world were presented and that a possible classification scheme was proposed.

We have rephrased the sentence: *"Then we establish a classification scheme of the landslide seismic signals from relevant signal features based on the analysis of the datasets of 13 sites."*

P8 L13-24: these lines sound more as part of introduction than data. The paragraph data should start from current L25 but a description of the analyzed dataset is actually missing: which is the length of the analyzed time series? How many events did you analyze? How did you select the events analyzed in the following paragraphs? I guess those were well-known events, or did you applied an automatic detection methods?

We added these information in the new version of the paper and in Table 2, section 4 and section 5 of the new version.

P8 L26: "For all sites, the instruments are deployed close to the landslide", what does "close" mean? Please be more specific. I guess that authors agree that, for example, two seismic sensors, one installed at 10 m and another one at 900 m from the very same landslide, would record signals pretty different, especially in terms of spectral content.

We added these informations in the description of the sites and in Table 2 and section 4. We mentioned that for each seismic network analyzed, at least one sensor is installed on the active zone or at its vicinity ($< 50$ m). Moreover, we choose to work with the most energetic trace for each recorded events that we assume to be the closest station to the source and hence, the most representative of the seismic sources properties.

Of course, the distance and the medium contributes in the features of the seismic signals and we do not decorrelate its contribution. But as mentioned in earlier answer, the source mechanism also contribute to the signals feature. We already justified our approach to limit the wave propagation influence (see earlier answers to the same comment). Basically, our assumption is (as mentioned in previous comment): different media and different propagation paths but same characteristics at different sites = source-controlled features.

P15 L14: "The signals present significant differences with the chosen features", please reword, the reader does not understand the meaning of this sentence.

We have rephrased the sentence.

P15 L15: "in the field, the differentiation", I am not sure to have correctly understood, maybe you meant "only from the seismic signal analysis, the difference between"?

OK. We have rephrased the sentence.

P17 L23: please avoid references that are not published work, i.e. Helmstetter

et al., (2017a), especially if the reference is used to support a very strong state-
ment such as "the high correlation between the repetitive events could only be
explained by stick-slip movement of the locked section(s)". A sentence like this
must be accompanied by supporting data or published results.
We removed the mentioned reference in this sentence.

P17 L29: concerning "most collapses occurred without precursory sequences
(Allstadt et al., 2017)", I would suggest tuning down this statement, which is
also in contradiction with P2 L24. There are a number of cases where precur-
sory seismic signals related to small rockfalls were documented, especially when
a station is installed nearby the slope or there is a local monitoring network. On
the contrary, when the closest station is distant or we do not dispose of other
monitoring data, recognizing those precursory events is difficult but potentially
there are. I also believe that the reference Allstadt et al. (2017) is not consistent
here.
OK. We agree and removed this sentence.

P18 L16-20: I do agree with "several descriptions of the seismic sources are pro-
posed for each study case" and a standard classification would help to discuss
and compare landslide-induced seismic signals. I understand that the authors
are proposing their classification as general reference, but I would suggest to the
authors to delete the sentence "we encourage future studies to use and possibly
enrich the proposed typology". In my opinion the scientific community does not
need to be encouraged to adopt one classification or another.
We disagree with this statement – standard typology does exist for instance
for volcano-related seismic sources or for glacier-related sources and have been
very useful to progress in the comparison of the seismic signals on all volca-
noes and in the creation of comparable catalogs. Though any standardiza-
tion/harmonization methods can be questionable, we believe that proposing a
nomenclature of sources is important for further discussions including rejecting
the proposed classification or interpretation.

By the way, why you do not adopt the classification proposed by Allstadt et al.
(2017)?
The classification proposed by Allstadt is not comparable to our classification
as it is related to detection and cataloging landslide failures at regional scales
($> 1$ km); the purpose of our classification is the slope scale.

P18 L25-27: reference needed
Done. We added the reference.

Reference list: the style is not consistent with the journal guidelines, in many
cases the doi is missing, there are repeated references (Hibert et al., 2014a),
others are missing (Provost et al., 2018) and there is some text here and there
probably out of place (e.g., P29 L10-11). An accurate revision of the reference
list is needed.

We corrected the style of the references taking into account the journal guidelines, and also updated the reference list.

Moreover, I do acknowledge the significant contribution of some of the authors to the field but I have the impression that self-citations are really abundant (five papers by Hibert et al., six by Helmstetter et al.). Please try to select your most significant works and refer to them.

We believe the citations related to the papers of Helmstetter and Hibert are relevant. We also added several new references to the manuscript from other research groups as proposed by reviewer # 1. We tried to be exhaustive in the references and we cite more than 130 papers (a significant number due to Table 1), in total around 18% of the citations are self-citations of the co-authors which we think is not over-abundant.

Figure 1: I would prefer the author to focus more on the sites from which they present some data instead of showing a collection of points in a global map. In addition, Figure 1 is redundant if one considers the list presented in Table 1.

We added a table gathering the informations about the analyzed sites and their seismic networks (Table 2). We removed Figure 1.

Table 1: some details/revisions are needed. 7 Alestch-Moosfluh: this landslide is also monitored with a geophone network (Manconi and Coviello, 2018); 8 Torgiovannetto, Assise: please modify in Assisi; 15 Aiguilles: Aiguilles Pas de l'Ours?; 22 US highway 50, CA: there is no reference/website about that?; 24 Millcoma Meander, Oregon: same as above; 33 Matterhorn peak/Mont Cervin: please use the international name (Matterhorn) or the Italian one (Cervino) and add the reference describing the more recent monitoring network (Occhiena et al., 2012); 48 Piton de la Fournaise caldeira: Piton de la Fournaise is not enough?; 53 Marderello torrent: the reference for this net- work is Coviello et al. (2015); 69 La Colima volcano: please use the international name (Colima Volcano) or the Mexican one (Volcán de Colima); 70 Merapi volcano flanks: please use Merapi volcano, be consistent with the list format; in addition, a number of sites are missing, especially overseas in USA (e.g., Kean et al., 2015), New Zeland (e.g., Lube et al., 2012), and South America (e.g., Kumagai et al., 2009; Worni et al., 2012).

OK. Thanks for providing this detailed information. We have corrected the Figure and Table 1 accordingly.

Figure 2: what about adding a sketch of the signal associated to each process?

We do not think this would had information at this stage of the paper. It seems to us that simple sketching cannot capture the complexity of seismic signals and that the representation we propose on figure 13 is more suited to expose this complexity.

Figure 13: I guess this is the most important figure of the paper, why does it only appear in the discussion?

This figure summarizes the presented signals properties. We do not think that an earlier presentation of this figure is necessary.

Given the large seismic dataset I suppose you have at your disposal, why did you plot only between 2 (most of the cases) and 6 (few cases) examples? I wonder if the variability of the attribute shapes is representative given limited number of examples here presented.

We present more examples in the discussion in the new version of the paper (Figure 12) and comment the variability of the attributes in the discussion (P17. l2-15).

**

**Reviewer 2: anonymous referee**

After reviewing the manuscript I read the review of Dr. Coviello. I fully agree with his comments. I will not repeat his comments in my review. I found in the manuscript some mistakes and problems. Accordingly, I propose a major revision of the paper (if not rejected in its present form), which is in line with my comments and those of V. Coviello. The document is verbose, with a lot of information (perhaps with little consistency in terms of content) and poor in conclusions. Please, be more concise and remove the unnecessary information. Justify the purpose of the paper better. References must be selected to shorten their number.

We thank the reviewer for these statements. The introduction of the paper has been thoroughly revised to better highlight the focus of the work. We also rephrased or deleted some sentences considered as verbose by the reviewer. All these changes are indicated in track mode changes in the revised version of the manuscript.

In general, I am very skeptical about the purpose of the paper: to establish a standard typology of endogenous seismic sources.

This comment has also been addressed by Reviewer #1. We believe that the compilation of case studies and the standardized processing and representation of the seismic events recorded on landslides we propose is relevant for the following reasons : 1) standardized classifications exist in other fields of microseismology such as in reservoir monitoring, slow earthquakes (LFE, VLFE, etc) and volcano monitoring; 2) Those standardized classifications have proven to be useful starting points for further discussions : the classification is never frozen and should evolve following new observations and models; 3) Compiling datasets from very diverse case studies allow to bring out the control of the source on the signals from each class (different media and different propagation paths but same signal characteristics at different sites = source-controlled features).

It is true that there has been a dramatic increase of monitoring/detecting seismic signals generated by different ground phenomena in the last five years. However, we have to bear in mind that seismic measurements are not a direct measure as they could be extensometers, for example. The terrain is so complex that I am skeptical about whether seismic monitoring could give detailed information about the phenomena.

We disagree with this statement. The arguments for using seismology, waveform analysis and analysis of the temporal and spatial distribution of seismic sources on landslides as complementary sources of information on the mechanics of the

processes are:

1) The temporal resolution of seismic instruments provides very accurate timing of the deformation processes and is non-invasive as it can detect events at distance from the sensor installation. These advantages are hardly met simultaneously with other types of sensor. Several studies have demonstrated the major contribution of seismology to built near-exhaustive catalogs of events at slope scale (Helmstetter et al, 2011; Dietze et al, 2017b), at regional scale (Hammer et al, 2016) and its potential for early-warning of debris-flows (Walter et al, 2017; Arratano et al., 1999; Burtin et al., 2009).

2) It records also the spatial distribution of the sources occurring in depth (Spillmann et al, 2007, Lacroix et al, 2011, Tang et al, 2015) which is not the case of extensometers for example. The location of the seismic activity represent valuable information to update geo-mechanical models determining the factor of safety of the slope (Tang et al., 2015).

3) The seismic signal features are controlled by the source mechanism providing insights in the mechanical behavior of the deformation.

4) Recent papers have also documented seismic signatures preceding the collapse of large landslides (Amittrano et al., 2005; Yamada et al., 2016; Poli 2017; Schöpa et al., 2018) proving the presence of seismic signals associated to slope instabilities deformation.

So, seismic data alone are very difficult to manage for mass movement studies, mainly if the signals are very short and are related to small energy release. We must be aware of the type of phenomena. In my opinion, it is the combination of different measurements that can contribute to information about the phenomena.

We agree and we never mentioned to consider seismology as a standalone technique for landslide monitoring.

For this paper, I suggest you only include the very significant seismic signals and avoid small events.

The purpose of the paper is clearly mentioned: we analyze the signals recorded at close distance to the slope ($< 1$km) with seismic sensor sensitive to the $\tilde{1}$-100 Hz frequency band. This means that we are exploring larger events and more distant events than Acoustic Emission studies (Dixon et al, 2015; Michlmayr, 2012) but smaller events than large slope failure (volume $> 1O^6$ m$^3$, Ekstrom and Stark, 2013). Obviously, the examples are "significant" signals at this scale as they are clearly above the noise level.

Mainly, because of the difficulty of a subsequent interpretation. This is in some ways one of the conclusions of the authors, given that they unify the "new named" slopequakes by including them all in one group. The slopequakes can be so complicated that the present catalogue is probably not complete.

We agree with the statement that slopequake signals can be "complicated" and "that the present catalogue is probably not complete". However, we also propose sub-classes taking into account the complexity of this class while keeping

a uniform denomination because they are usually analyzed as one class in the previous and current studies. The name "slopequake" was chosen in order to remove the source mechanisms interpretation induced by the name "slidequake" or "micro-earthquake". As mentioned earlier, the present classification is not frozen and can be enriched and/or discussed. In particular, for certain sub-classes we explicitly mentioned that surface processes may also generate these type of signals (SQ-tremor like signals and SQ-with precursory).

All efforts in this line should be devoted to monitoring one site with different instruments and to interpreting the events in coordination with different specialists.
This is done on most of the recent sites being instrumented by different research groups. However, the source mechanisms and the variability of the slopequake features remain poorly documented. Understanding this variability and the underlying physical processes remains a strong challenge, and we hope that the classification we propose will bring some insights leading toward a better grasp of those processes.

*Having said this, see below for further comments.*

1) Table 1. This is a very risky approach. Given the present increase in mass monitored studies you probably miss one unless it is the intention of the authors only to mention those in which they are involved. In this case, you must mention this specifically, and give your reasons.
We agree that we probably missed some sites especially the new sites recently instrumented. We added the missed sites suggested by V. Coviello (4 over 70). If we are missing further references, please, let us know and we will add them to the table.

2) As regards field instrumentation, it would be useful to better explain the characteristics of the instruments and their different site conditions. Site effects are completely ignored in the interpretation/description of the signals. In fact, most of the presented data were already the subject of different interpretations and I assume that these have been described in the corresponding papers. However, when seeking to establish a standard typology, consideration of the peculiarities (or not) of the site effects is very important.
This comment was already addressed by Reviewer #1.
Concerning the field instrumentation, we propose a new section "Data" (section 4) to describe precisely the seismic network configuration (also summarized in table 2). The geomorphological and geological context are indicated in Table 1 with the references for further information.
Concerning the instruments, we corrected the instrument response and analyzed their common frequency band.
Concerning the site effects, it is true that we do not correct. However, we believe that the comparison of signals from different sites of various geomorphological and geological contexts is precisely a good strategy to discriminate the contribution of the source mechanism from the site effects/attenuation. We hence describe the features shared by all the selected examples without focusing on particular features of certain signals that are likely linked to site effects.

3) In the definition of the parameters of processing methodology, if I am not mistaken, no amplitudes are considered (only once on pag. 15 line 21). Why are the amplitudes (nm/s) not indicated in the events? It is true that attenuation can also affect amplitudes, not only the frequency content, but it could be useful for differentiating events. The relative "energy" released together with the duration of the signals can give significance to some events.

Amplitudes are indicated on the figures for each trace of each example in nm/s. We did not choose to analyze the amplitudes or Energy/duration relationships (even so, they can be significantly different from one class to the other) because we are focusing in the features that can be related to the source mechanisms and not to its magnitude.

4) Additionally, the authors devote a large description to the frequency content of the events. However, in figure 13, the maximum attention is devoted to other parameters that are basically in the time domain.

Over 9 parameters presented on figure 13, 5 are related to the frequency content and 4 to the time domain. The waveforms presented on figure 13 gather information on both time and frequency content. Moreover, we think that the format of the figure used to present examples for each classes (Figures 3 to 12) summarizes all the informations needed to discuss the signals. On Figure 13, we adopted "star" diagrams in order to ease the visualization of all the selected features and not to focus only on frequency nor time domain properties.

Moreover, the parameters introduced in the processing methodology section are not sufficiently considered in the description of the events. Note that in the description, few of these parameters are mentioned.

OK. We reviewed these sections to add these informations.

Furthermore, the last sentence of section 4 merits a detailed explanation and challenges the classification. In this sentence, the authors mention the real problem of the dependence of the defined parameters on the source to sensor distances and on the propagation media properties.

OK. We have rewritten this part of section 4 (Section 5. in the revised manuscript) to describe our approach to analyze the datasets and compare them (P10. l21-30).

5) Pag. 11. Explanations and description of the signals are very poor. Some explanations correspond to other cases. I include my comments about the case of RF (pag. 11) only as an example.

We reviewed these sections to add further informations.

Pag. 11 Line 2. Please, indicate if the rockfall was monitored. Information

specific on this event is necessary.

OK. We added information about this event on the description of the datasets.

Line 5. What does it means:. energy below 10 Hz is present for volume larger than 1 m$^3$ (Fig 3a). Is this your case, because you mention this figure here?

It is not only the case of this specific event. We added references to support this statement in the next paragraph: *"The frequency content is also controlled by the block mass i.e. the frequency of spectral maximum energy decreases when the block mass increases (Farin et al, 2015; Burtin et al, 2016; Huang et al, 2007"*.

Line 7. The study of Farin et al., (2014) is an experiment in lab. and cannot be extrapolated to nature as it is observed (the high freq. disappear). This contribution is no relevant here.

OK. We removed this statement.

Line 9. P- and S- waves are hardly distinguishable. Is this in this specific case? You cannot generalize.

The statement is supported by different references.

Line 11. First arrivals are mainly impulsive. At the scale of representation I have to believe it.

We removed statements concerning the impulsive nature of the signals as it may vary from site to site.

Line 12. Figure 4 is incomplete. Information is required. If the signal belongs to a publication, the references must be included. Otherwise a comment is necessary.

The reference of the dataset from which the presented events are taken from are added in the caption when the presented (or similar events) have been published.

Line 13. Why do you suspect that the signals could be different if what you are recording is the movement of the mass falling down the slope? Normally, there is a time lag between ground and blast signals and the signals of the rock fall as observed in earlier publications.

We meant that natural rockfall are often composed of several falling blocks subject to break-up. In the case of the Riou-Bourdoux experiment only single block falls were monitored. It is true that in other studies when the rockfall is triggered from the rock cliff, very similar mechanisms and signals can be observed. We hence rephrased the sentence accordingly.

Lines. 18 and 20. Le Roy et al. 2017and 2018. Complete appropriately these references

OK. We have corrected the references.

Line 22. Burtin et al, 2016. This paper is devoted to torrential process. By the

way, the reasoning in the outlook section is of interest.

Fig1. of Burtin et al, 2016 shows the influence of the block mass on the frequency content even so, it is not discussed in the text. We added Huang et al, 2007 that discusses the same experiment.

Line 24. You mention "[...] may be emergent due to simultaneous arrivals of the waves". Explain this better. Do you mean that it could be interference between the impulses? What happens with the wave field? It also depends of the frequency content.

OK. We rephrased the sentence as: " [...] may be emergent due to simultaneous arrivals of waves generated by impactors of different sizes impacting the ground at closely spaced time intervals".

6) Figure 13 is perhaps one of most interesting figures but it must be better explained. As I mentioned before, small events must be avoided.

We enriched the discussion of this figure (P.16 l34 to P.17 l.7). We already respond to the "small events" issue in a previous comment. Basically, we selected events clearly above the noise level.

7) Pag. 16 Line 3-4. The authors justify the differences in the frequency content mentioning attenuation because of large distances, but this is not the case here because it is indicated in the paper and in the abstract that the signals are from events at r< 1 km. Is this consistent?

Attenuation is function of the distance and the wavelength of the seismic waves observed. c.f. previous response to comments of Reviewer #1. At our scale, "large distances" ranges from 100 m and more, depending on the magnitude of the source and the network geometry. One can clearly observed the influence of the distance in most the presented examples (Figures 3 to 11), even if the location of the source is not computed. Moreover, this comment is in contradiction with all the previous comments concerning the influence of the wave propagation on the recorded signals as a strong limitation of our study.

8) The term seismologically is not used correctly in the text. Replace it by seismic instruments. What does seismologically instrumented mean? In the world of seismologists the instruments are seismic instruments or not. They could have different resolution, characteristics, etc: : : "Seismologically" refers to a discipline, but not to the installation. Basically, the parameters you are considering are devoted to data processing signals and signal characteristics and not to wave transmission which is the subject of seismology. And as regards the installation, what does a non-seismologically installed seismic instrument mean?

OK. We corrected accordingly.

9) Pag. 16 line 19 and below. All this information devoted to harmonic signals in the discussion section is out of place. Moreover, it does not correspond to the data presented by you.

We present data from our gathered dataset. Except the one recorded at the Slumgullion landslide (Gomberg et al.), none of these signals have been published before. We discuss why we do not refer to these signals in the proposed classification.

10) Discussion. From line 19 to the end. The information provided does not correspond to a discussion of what is presented in the paper. It mainly concerns previous results without comparing them with the data presented in the manuscript.
We have thoroughly rewritten the discussion section.

Most of the sentences in the discussion could be included in the introduction, because the information is previous to the results presented in the paper and with little relation to them, at least in the present form.
We have rewritten the introduction and the discussion to take into account this comment and the previous ones.

Moreover, I do not understand why the harmonic signals are included in the discussion.
We discussed the harmonic signals in the Discussion section as we are not including them in the proposed classification whereas they have been presented in other studies. We find surprising that, on the one hand, reviewers reproach us not to compare our data to other and then, on the other hand, find the paragraph where we do this comparison not relevant.

11) As the authors mention on pag. 4 citing (Walter et al., 2017) in MS processing chains (by the way, I do not understand why you include this information in this section): "Many studies approximate the media attenuation field and/or the ground velocity, or do not take into account the topography, leading to mislocation of the events that prevents for accurate interpretation of certain sources and leads to false alarms". Is this the case of the data presented here?
We talked about location of the source which is an important information to associate the recorded signals to slope deformation. However, we mentioned here that location using attenuation law and assuming a homogeneous attenuation factor may lead to mislocation of the seismic events. Consequently, if the location error is of the same order of the distribution of geomorphological structures, it can be difficult to interpret the source of the recorded signals.
In the present study we did not locate the events and focus on the signals features that can be related to the seismic source mechanism. The later is discussed in each sub-classes presentation with reference to studies that modeled the seismic sources from the seismic signals or to studies that observe similar signals in different context (e.g. glacier motion).

12) Papers under revision although they are public must not be cited, nor must papers in volumes without a standard scientific recognition.
We cite posters and abstracts only to present the monitoring sites and/or the

datasets (Table 1 and Figure 2 to 10). We removed reference to posters/abstract when supporting statements in the text. For the papers under revision, we let the editor decides whether they should be included in the reference list (most of them being today accepted).

Some comments on the analyses of data. 13) As regards the tremor-like slope-quake (you do not mention this in this way in the title of figure caption of Fig. 12), the PSD is in the range of 8-13Hz, (not 10 Hz as mentioned) and the mean frequency of 20Hz is not clearly deduced from the plots.
OK. We corrected the description of this class and the caption.

14) Slope-quake with harmonic coda (H-SQ). I do not only observe the coda in the Chamousset signal (fig.10a) (note there is an error in this figure) of 08 August, but also in that of 6 October (fig. 11c). Super-Sauze site slope-quake signal of 24 Oct. (Fig. 10b) and the rock fall signal of 5 Nov (Fig. 4d) also present this behavior. This harmonic coda is present in different events. I think this is significant, and perhaps this is not related to the source but to the site effect for specific frequencies.
We agree that for this particular case, wave propagation could be a better explanation for the signal feature. Consequently, we removed this class from the classification and we discuss this signal feature in the new version of the discussion.

15) As regards all figures, but specifically Fig. 3 since it is the reference. Please, indicate the information contained in the plots in the figure caption. What is Amax? Is the parameter defined in section 4? It could be informative to show the maximum amplitude in ground speed units. What are the different traces in different colors shown in plot a? Indicate correctly the power of 10 (10 ˆx and not eˆx).
OK. We indicated that $A_{max}$ refers to the maximum amplitude (nm/s). The different traces in different colors correspond to the other sensors present on the site, we added this comment in the caption. We modify the power accordingly.

[revised manuscript text omitted]

– **Monochromatic slopequake (Mono-SQ)** (Figure ??): The first type of Complex Slopequake signals present an almost

10 monochromatic frequency content with no harmonic. The signals are almost symmetrical and no fracture event is observed at the beginning which differentiate them from Mix-SQ. Conversely to the LF-SQ, their frequency bandwidth is narrow. In the case of Slumgullion (Figure ??.b), 90 repeaters of this event were measured during the month of observation (?). The fundamental frequency is 11.9 Hz with a standard deviation of 0.7 Hz computed from the stack of the signals with a correlation coefficient higher than 0.7 (?). The authors argued that the resonance is a property of the

15 source considering the stability of the fundamental frequency through time and the absence of anthropogenic sources in the vicinity of the landslide. They hypothesize that the waves were trapped along the side-bounding strike-slip fault generated by shear events. The location of the source, the distribution of the amplitude, the stability of the fundamental frequency and the daily temporal occurrence of the source supports this assumption. Similar kind of events occur at the Super-Sauze landslide.

[revised manuscript text omitted]

| Number | Site | Location | Type | Material | Sensor | Duration | Reference/Research Group |
|--------|------|----------|------|----------|--------|----------|--------------------------|
| 1 | Randa | Switzerland | Slide | Hard rock | G | SC |  ? |
| **2** | **Séchilienne** | France | Slide | Hard rock | G, SP | P |  ??? |
| **3** | **La  Clapière** | France | Slide | Hard rock | SP(?) | P |  ?? |
| **4** | **Aaknes** | Norway | Slide | Hard rock | G,BB | P |  ? |
| 5 | Peschiera Spring | Italy | Slide | Hard rock | A | SC |  ? |
| 6 | Gradenbach | Austria | Slide | Hard rock | SP | P(?) |  ? |
| 7 | Alestch-Moosfluh | Switzerland | Slide | Hard rock | BB | P |  ?? |
| 8 | Torgiovannetto,  Assisi | Italy | Slide | Hard rock | SP | SC |  ? |
| | | | Continued on next page | | | | |

| Number | Site | Location | Type | Material | Sensor | Duration | Reference/Research Group |
|---|---|---|---|---|---|---|---|
| 9 |  Akatami landslide | Japan | Slide | Hard rock | (?) | (?) |  |
|  10 | Akkeshi landslide | Japan | Slide | Hard rock | SP | P |  |
|  11 | Rausu landslide | Japan | Slide | Hard rock | BB | P |  |
|  12 | Fergurson slide / Mercel River | USA / California | Slide | Hard rock | (?) | (?) |  |
|  13 | Turtle Mountain - Frank slide | Canada | Slide | Hard rock | G | P |  |
|  14 |  Aiguilles-Pas de l'Ours | France | Slide | Soft rock / Earth | BB | SC |  |
| 15 | Harmalière | France | Slide | Soft-rock | SP,BB | P | ? |
| 16 | Utiku | New Zealand | Slide | Soft rock / Earth | (?) | P |  |
| 17 | Villerville | France | Slide | Soft rock / Mud | BB | SC,P |  |
| **18** | **Super-Sauze** | France | Slide | Soft rock / Mud | SP | P, RC |  ???? |
| **19** | **Pont Bourquin** | Switzerland | Slide | Mud | SP(?) | P |  ?? |
| 20 | Valoria | Italy | Slide | Mud | SP | SC |  |
| **21** | **Pechgraben** | Austria | Slide | Mud | SP,BB | RC |  |
| 22 | US highway 50, CA | USA | Slide | Earth | G | P | USGS (https://landslides.usgs.gov/monitoring/) |
| **23** | **Slumgullion** | USA | Slide | Earth | G | RC |  ?? |
| 24 | Millcoma Meander, Oregon | USA | Slide | Earth | G | P | USGS (https://landslides.usgs.gov/monitoring/) |
| 25 | Xishancun | China | Slide | Earth | BB | SC | - |
| 26 | Chambon Tunnel | France | Slide | Earth | SP | P | - |
| 27 | Maca | Peru | Slide | Soft rock / Earth | SP | P(?) |  |
| 28 | Heumoes | Germany | Slide | Soft rock / Earth | SP | RC |  |
| 29 | Mission Peak landslide | USA / California | Slide | Soft rock / Earth | BB | P |  |
| 30 | Char d'Osset | France | Slide, Fall | Soft rock / Mud | | | - |
| 31 | Mesnil-Val | France | Fall | Hard rock | G | SC |  ?? |
| 32 | North Yorkshire coast | United Kingdom | Fall | Hard rock | BB | P |  |
| 33 | Matterhorn peak  | Italy | Fall | Hard rock | G | RC |  ?? |
| 34 | Madonna del sasso | Italy | Fall | Hard rock | SP | P(?) |  |
| **35** | **Chamousset** | France | Fall | Hard rock | SH | RC |  ?? |
| 36 | Mont-Granier | France | Fall | Hard rock | BB | P | - |
| 37 | Les Arches | France | Fall | Hard rock | SP | P(?) |  ?? |
| 38 | La Praz | France | Fall | Hard rock | SP | P(?) |  |
| 39 | Rubi | France | Fall | Hard rock | SP | P(?) |  |
| 40 | La Suche | Switzerland | Fall | Hard rock | SP | P(?) |  |
| **41** | **St. Eynard** | France | Fall | Hard rock | SP | P(?) |  ?? |
| 42 | Cap d'Ailly | France | Fall | Hard rock | | | - |
| 43 | Lauterbrunnen valley | Switzerland | Fall | Hard rock | BB | SC |  ?? |
| 44 | Three Brothers | USA | Fall | Hard rock | SP | SC |  |
| 45 | Mount Néron | France | Fall (triggered) | Hard rock | BB | SC |  |
| **46** | **Riou Bourdoux** | France | Fall (triggered) | Hard rock | SP,BB | SC |  |
| 47 | Montserrat | Spain | Fall (triggered) | Hard rock | SP | SC |  |
| **48** | **Piton de la Fournaise** | France | Fall, Flow | Volcanic rock | BB | P | OPVF/IPGP,  ???? |
| 49 | Bolungavík - Oshlíðslope | Iceland | Fall, Flow | Hard rock | A | P |  |
| **50** | **Rebaixader** | Spain | Flow | Debris | G | P |  ???? |
| 51 | Manival torrent | France | Flow | Debris | G | P |  |
| 52 | Réal torrent | France | Flow | Debris | G | P |  |
| 53 | Marderello torrent | Italy | Flow | Debris | G | P |  |
| 54 | Acquabona torrent | Italy | Flow | Debris | G | P(?) |  ?? |
| 55 | Moscardo torrent | Italy | Flow | Debris | SP | P |  |
| 56 | Gadria torrent | Italy | Flow | Debris | G | P |  |
| 57 | Mt. Yakedake volcano - Kamikamihorizawa Creek | Japan | Flow | Debris | | |  |
| 58 | Lattenbach torrent | Austria | Flow | Debris | G | P(?) |  ?? |
| 59 | Illgraben torrent | Switzerland | Flow | Debris | G | P |  ?? |
| 60 | Farstrine torrent | Austria | Flow | Debris | G | P(?) |  |
| 61 | Wartschenbach torrent | Austria | Flow | Debris | G | P(?) |  |
| 62 | Dristenau torrent | Austria | Flow | Debris | G | P(?) |  |
| 63 | Shenmu creek | Taiwan | Flow | Debris | G | P |  |
| 64 | Ai-Yu-Zi creek | Taiwan | Flow | Debris | G | P |  |
| 65 | Fong-Ciou creek | Taiwan | Flow | Debris | G | P |  |
| 66 | Chenyoulan creek | Taiwan | Flow | Debris | G | SC |  |
| 67 | Mt. Sakurajima Volcano - Nojiri Torrent | Japan | Flow | Debris | G | P |  |
| 68 | Mount Pinatubo | Philippines | Flow | Debris | G | P |  |

[revised manuscript text omitted]

---

## Referee Report (RR1)

**Review comments**

esurf-2018-23-manuscript

[Towards a standard typology of endogenous landslide seismic sources]

August 18, 2018

The manuscript presents a standard typology of seismic sources including slopequake, rockfall, and granular flow based on the features of seismic signal and characteristics of its frequency content, leading a possibility of the forecast of initiation of landslide rupture. The revised version of the manuscript significantly improved in overall quality and I think that the subject is relevant to publication in Earth Surface Dynamics. However, there are several places where I think a bit more explanation and minor-to-moderate revision are needed. Detailed comments are listed below.

**Abstract**

(1) P1., Lines 6-8 "…We investigate the 1-100 Hz frequency band…":

The signal features (e.g., duration, the dissymmetry coefficient, the number of peaks of the envelop function) used in the typology analysis highly depended the filtered seismic signals. How changes in the pattern of the proposed classification shown in Fig. 11 if the authors use a bandpass filtering with different frequency range, for example of 1-50 Hz. I also found the claim of the authors "The signal features are always computed on the trace with the maximal amplitude band-passed in the range [$f_c$-50] Hz" (P12., Lines 4-5). Is there a mismatching compared to the 1-100 Hz stated in the abstract? The frequency content of a few source events can be higher than 50 Hz, such as resulting shown in Fig. 2(a), Fig. 3(c,e), Fig. 5(a,b), Fig. 7(e). If the authors applied a band-pass filtering of $f_c$-50 Hz, it would be an unfair comparison in the classification scheme using signals recorded at different sites from variable sources. The author need to clarify above discrepancy.

**1. Introduction**

(2) P3., Line 2 "Schöpa et al., 2017":

Please replace the reference of Schöpa et al. (2017) by "Schöpa et al. (2018) Dynamics of the Askja Caldera July 2014 landslide from seismic signal analysis: precursor, motion and aftermath, *Earth Surf. Dyn.,* 6, 467–485, https://doi.org/10.5194/esurf-6-467-2018" in the manuscript.

**3.3 MS processing chains**

(3) P9., Lines 9-10 "…For this kind of signal, location methods based on the inter-trace correlation of the surface waves (Lacroix and Helmstetter, 2011)…"

A location scheme based on the cross-correlograms of inter-stations was proposed by

Chen et al. (2013) for the determination of landslide source location. Please add the reference of "Chen et al. (2013) A Seismological Study of Landquakes Using a Real-Time Broadband Seismic Network. *Geophys. J. Int.*, 194, 885-898, doi:10.1093/gji/ggt121."

**5. Methodology**
(4) P11., eq.(2):
If I understand correctly the $A_{max}$ should be replaced by "$T_{corr}$". The value of $A_{max}$ is the maximal amplitudes of the waveform trace, which defined in the caption of Fig. 2.

**6. Seismic description of the signals - typology**
(5) P12-P16:
Why do the authors use the vertical trace in typology analysis only? Maybe some of events were recorded by vertical geophone. Did you try to use the horizontal components that generally less noisy? The seismic phases excited by the sliding stage are likely composed primarily of Rayleigh wave or shear wave, which have relative large seismic energy in the horizontal components. Considering both vertical and horizontal signals also helps us to comprehensively understand the possible source type in seismic wave generation. For example, the impacts of rockfall can excite higher seismic amplitudes in the vertical component.

**6.3.2 Complex Slopequake**
(6) P15., Lines 29-33:
Is the small-sized deeper earthquake (focal depth is relatively larger than inter-station distance) possible to be a seismic source of precursor signals? In cases of Fig. 9 a,b, the amplitude of precursor signals exhibits no obvious decay at different station sites. The amplitude ratio of vertical to horizontal for local deeper earthquake would be higher due to high incident angle of ray path of seismic waves. Three-component traces to discuss it would be helpful I think.

(7) P16., Lines 18-20, 31-32; P18., Lines 5-7; P19., Lines 31-32:
I also found the claim of the authors further investigations of precise source location and/or source mechanisms are needed to improve understanding different sources occurred at unstable slopes. However, the authors did not discuss more details in providing possible solutions to solve location and mechanisms of sources in accuracy. They (P16., Lines 18-20, 31-32; P18., Lines 5-7; P19., Lines 31-32) are weak statements in the current manuscript.

**7. Discussion**

(8) P17., Lines 29-30 "Harmonic signals have been also been documented recorded at the Pechgraben and Super-Sauze landslides Vouillamoz et al. (2017)":

The sentence needs some adjustments regarding the English grammar, please make it more easily readable.

(9) P18., Line 8 "Harmonic coda are also observe for certain signals (Fig. 3d, Fig. 9c) at high frequencies …":

It is not clear to me. Please highlight the portions of harmonic coda in the figures.

**Figure 11.**

(10) What is the meaning of n-value shown in upper right corner of each radar chart? Is n-value the number of used traces in classification analysis? Please clarify it.

---

## Author Response (AR2)

Dear Editor,

We thank warmly Dr. Velio Coviello, Wei-An Chao and the editor for their review of the manuscript and their constructive comments. We propose below detailed answers, thoughts and clarification concerning the main points of interrogations of referees.

Sincerely,
Floriane Provost on behalf of all co-authors,

NOTE: In the following document, the referee comments are in normal fonts and the answers are in blue font.

**

**Reviewer 1: Dr. Velio Coviello**

I have read the point-to-point reply prepared by Provost and colleagues. I think the authors have done a good job of responding to the review comments. I appreciated their thorough justifications and the changes they made to the paper. There are no new major comments to the manuscript from my part. As such, I recommend the paper for publication after minor revisions. Please find here following my main comments to the revised manuscript. My final recommendations are not compulsory, authors (and the editor) will decide if accepting them or not, let me know what do you think.

1) The authors strongly support the use of the spectral information for the classification of landslide-induced signals detected at a distance ¡ 1 km. In particular, they argue that "the source mechanism is a predominant factor controlling signal spectral content although the sensor to source distance plays a role in the contribution of certain frequencies to the signals amplitude". This is probably ok but, in my opinion, the possible limitations of this approach deserve to be mentioned, especially if the ambition of the paper is to propose a general classification. We know that source proprieties have a strong control on the signal characteristics, this is self-evident. We also know that propagation through the geological media acts as a low-pass filter. However, not only the source-sensor distance but also attenuation and resonance effects can play an important role in shaping the spectral response. Imagine a number of sensors installed at the same distance from the source but in different conditions. The installation methods (directly in the ground, on a boulder, in a case placed on compacted deposits, etc.) and the geological context (sensor installed on bedrock, on fine deposits, on the landslide body, etc.) have a strong effect on both amplitude and frequency domains, in particular in case of low-cost and temporary monitoring networks. I think this can have an impact not only in terms of location error but also on the source characterization. In my previous

review I reported some examples of these local effects, but there are many others published.

We added a sentence in the discussion: *"Difficulties still arise in providing an exhaustive description and interpretation of all the sources from the simple analysis of the proposed signal features, particularly those generating short-duration signals. The ambiguity between propagation effect and source mechanisms prevent further interpretation of the classes discriminated by the frequency content such as LF-SQ and HF-SQ. Several limitations currently prevent to invert this kind of sources with accuracy."*

2) In their response about type of sensors, network geometry, and local effects, the authors summarize their thoughts with this statement: "Regarding the geological and geomorphological contexts, our assumption is that if we can observe similar signal features in different sites they can only be explained by the similarity of seismic sources". Similarity is a classical and elegant attribute employed in many classifications of natural phenomena. Keep focusing on landslides, similarity is the baseline of the most adopted classification scheme proposed by Cruden and Varnes (1996), i.e. "similar landslides in similar materials are caused by similar processes under similar conditions". However, I think that such a simple criterion could be in some cases misleading when the classification is only based on indirect measurements of the object of study, i.e. the inversion of landslide-induced seismic signals.

We adopted this criterion as it is commonly used in seismology to establish classification of seismic signals generated by tectonic faults, volcanoes or glaciers.

Minor corrections: the reference list still needs a careful review, still there are some contradictions on the number of test sites (13 or 14?),

There are 13 sites, we corrected the number accordingly.

Table 1 still needs some correction (please refers to previous review round comments),

We carefully corrected table 1 according to the previous comments. We should not miss any of the corrections now.

the unit of measurement is missing in all seismographs, the bar scale is missing in all spectrograms.

The units of the measurments are indicated in figure caption. Scales are added for all spectrograms.

Cruden DM, Varnes DJ (1996) Landslide types and processes. In: Turner AK, Schuster RL (eds) Landslides investigation and mitigation. Transportation research board, US National Research Council. Special Report 247, Washington, DC, Chapter 3, pp. 36–75.

**Reviewer 2: Dr. Wei-An Chao**

The manuscript presents a standard typology of seismic sources including slope-quake, rockfall, and granular flow based on the features of seismic signal and characteristics of its frequency content, leading a possibility of the forecast of initiation of landslide rupture. The revised version of the manuscript significantly improved in overall quality and I think that the subject is relevant to publication in Earth Surface Dynamics. However, there are several places where I think a bit more explanation and minor-to-moderate revision are needed. Detailed comments are listed below.

Abstract
(1) P1., Lines 6-8 "...We investigate the 1-100 Hz frequency band...": The signal features (e.g., duration, the dissymmetry coefficient, the number of peaks of the envelop function) used in the typology analysis highly depended the filtered seismic signals. How changes in the pattern of the proposed classification shown in Fig. 11 if the authors use a bandpass filtering with different frequency range, for example of 1-50 Hz. I also found the claim of the authors "The signal features are always computed on the trace with the maximal amplitude band-passed in the range [fc-50] Hz" (P12., Lines 4-5). Is there a mismatching compared to the 1-100 Hz stated in the abstract? The frequency content of a few source events can be higher than 50 Hz, such as resulting shown in Fig. 2(a), Fig. 3(c,e), Fig. 5(a,b), Fig. 7(e). If the authors applied a band-pass filtering of fc-50 Hz, it would be an unfair comparison in the classification scheme using signals recorded at different sites from variable sources. The author need to clarify above discrepancy.

The signals are filtered in the fc-50 Hz band when computing the different features. Higher frequencies are only displayed in the figures presenting examples of signals. This band is chosen because it is the common frequency band of the different instruments installed in the different study sites. For clarity, we now mentioned the 1-50 Hz band in the abstract.

1. Introduction
(2) P3., Line 2 "Schöpa et al., 2017": Please replace the reference of Schöpa et al. (2017) by "Schöpa et al. (2018) Dynamics of the Askja Caldera July 2014 landslide from seismic signal analysis: precursor, motion and aftermath, Earth Surf. Dyn., 6, 467–485, https://doi.org/10.5194/esurf-6-467-2018" in the manuscript.

We corrected accordingly.

3.3 MS processing chains
(3) P9., Lines 9-10 "...For this kind of signal, location methods based on the inter-trace correlation of the surface waves (Lacroix and Helmstetter, 2011)..."
A location scheme based on the cross-correlograms of inter-stations was proposed by Chen et al. (2013) for the determination of landslide source location. Please add the reference of "Chen et al. (2013) A Seismological Study of Landquakes Using a Real-Time Broadband Seismic Network. Geophys. J. Int., 194, 885-898, doi:10.1093/gji/ggt121."

We do not agree to add this reference. The mentioned paper proposed a methodology to locate landquakes being recorded by a regional seismic network. We choose to only mentioned studies that attempt to locate signals of much smaller energy only recordable at closer distances than the ones presented in Chen et al, 2013.

5. Methodology
(4) P11., eq.(2): If I understand correctly the Amax should be replaced by "Tcorr". The value of Amax is the maximal amplitudes of the waveform trace, which defined in the caption of Fig. 2.
Yes, indeed. We corrected accordingly.

6. Seismic description of the signals - typology
(5) P12-P16: Why do the authors use the vertical trace in typology analysis only? Maybe some of events were recorded by vertical geophone. Did you try to use the horizontal components that generally less noisy? The seismic phases excited by the sliding stage are likely composed primarily of Rayleigh wave or shear wave, which have relative large seismic energy in the horizontal components. Considering both vertical and horizontal signals also helps us to comprehensively understand the possible source type in seismic wave generation. For example, the impacts of rockfall can excite higher seismic amplitudes in the vertical component.

We presented vertical components because the majority of the seismic network sensors are vertical component sensor. Horizontal traces are not systematically available and analyzed.

6.3.2 Complex Slopequake
(6) P15., Lines 29-33: Is the small-sized deeper earthquake (focal depth is relatively larger than inter-station distance) possible to be a seismic source of precursor signals? In cases of Fig. 9 a,b, the amplitude of precursor signals exhibits no obvious decay at different station sites. The amplitude ratio of vertical to horizontal for local deeper earthquake would be higher due to high incident angle of ray path of seismic waves. Three-component traces to discuss it would be helpful I think.

We don't think a small-sized deeper earthquake may be a possible source for the precursor signals. As mentioned in the text, the amplitude of the most energetic part of the signal and of the precursor signal presents a decay at different station sites that makes the interpretation of a deeper earthquake unlikely.

*"At all sites, the amplitude of the signal is significantly higher for one of the sensor (3 to 50 times higher) when considering vertical traces. The precursory signal is buried in the noise at the sensors with lowest amplitudes and the signal is similar to a LF-slopequake."*.

In the case of Super-Sauze, one of the 3C sensors records the precursory signal. The traces and the ratio of vertical over horizontal amplitudes are shown in the figure below (Figure 1 and 2). The precursor signal are well observed for one of the two 3C sensor (Figure 1). In this case, the amplitude ratio of vertical to horizontal increases slightly at the beginning of the precursory signals but vertical amplitudes are lower than horizontal amplitudes. Vertical amplitudes become larger than horizontal amplitudes at the beginning of the most energetic part of the signal that could be coherent with P-wave arrivals generated to the block impact. The latter is also observed at the other sensor (Figure 2).
We agree that a systematic installation is needed for a better comprehension of the source. We added a comment in the discussion and in the conclusion.

(7) P16., Lines 18-20, 31-32; P18., Lines 5-7; P19., Lines 31-32: I also found the claim of the authors further investigations of precise source location and/or source mechanisms are needed to improve understanding different sources occurred at unstable slopes. However, the authors did not discuss more details in providing possible solutions to solve location and mechanisms of sources in accuracy. They (P16., Lines 18-20, 31-32; P18., Lines 5-7; P19., Lines 31-32) are weak statements in the current manuscript.
We discuss the current limitations in the discussion (P18 L20 -P19 L10). As indicated on the discussion, we think that the main efforts must be, first, made to improve the seismic networks and construct large and exhaustive catalogs to investigate temporal and spatial occurrence of the sources. We changed slightly some of the mentioned statements (see P18 L5-7, P19 L31-32).

7. Discussion
(8) P17., Lines 29-30 "Harmonic signals have been also been documented recorded at the Pechgraben and Super-Sauze landslides Vouillamoz et al. (2017)": The sentence needs some adjustments regarding the English grammar, please make it more easily readable.
We corrected the sentence.

(9) P18., Line 8 "Harmonic coda are also observe for certain signals (Fig. 3d, Fig. 9c) at high frequencies ...": It is not clear to me. Please highlight the portions of harmonic coda in the figures.
The spectrogram of the coda shows harmonic frequencies. We reformulated the sentence:
*"For certain signals, the coda is dominated by resonance frequencies (Fig 3d, Fig 9c) at high frequencies (i.e. 20 and 43Hz), well observed in the spectrogram of the signal."*

Figure 11.
(10) What is the meaning of n-value shown in upper right corner of each radar chart? Is n-value the number of used traces in classification analysis? Please clarify it.
Yes it is. We completed the caption.

[Figure]

Figure 1: The three upper subplot display the Vertical and horizontal seismic traces for the Complex-Slopequake with precursor of the Super-Sauze landslide (12-Nov-2014 00:36:59, Seismic array A). The lower subplot shows the amplitude ratio between vertical and horizontal traces. The precursor signals are starting approximately at 1s.

[Figure]

Figure 2: Same as Figure 1 for Seismic array B.

**

**Editor corrections:**

Some minor language points:

5.2 thousands of cubic meters

5.12 triggers

7.27 citation in parentheses

9.21 is the citation of Cruden (1996) different to the one of Cruden and Varnes (1996). If yes, one citation is sufficient. If not, please include 'see also' or similar into the parentheses.

12.18 launched

15.29 The second class... has the same general properties as...

15.10 Humans could also be active at night. Tone this down by saying '...makes a human source unlikely...'

17.2 ...consisted of...

All the above sentences have been corrected accordingly. The citation of Cruden and Varnes 1996 is corrected.

[revised manuscript text omitted]

---

## Author Response (AR3)

Dear Editor,

We thank warmly the editor for its careful review. We corrected the grammatical and spelling mistakes and rewrote some sentences of the discussion. We hope this

Sincerely,
Floriane Provost on behalf of all co-authors,

NOTE: In the following document, the referee comments are in normal fonts and the answers are in blue font.

**

**Editor comments:**

Dear authors,
thank you for the revised paper. I find that the reviewers' comments are adequately addressed. However, there are a large number of small language problems. These fall in two categories: first, typos and small grammatical errors - the list below is probably not exhaustive, so please check carefully through the manuscript. Second, there are some awkward and unclear phrases remaining, especially in the discussion. Please strive to improve these.
Note that the abstract conventionally is a summary of the article and as such is not part of the paper. Please make sure that abstract and manuscript can be read independently of each other. Currently, some information (e.g., used frequency bands, in response to a review comment) can only be found in the abstract. Please rewrite this accordingly and make sure that the abstract actually gives a summary of the most important results. As an example, it should give a brief description of the different classes you propose. Methodological detail is not necessary in the abstract.
I am looking forward to seeing your revised manuscript,
with thanks and best wishes, Jens Turowski

Abstract: the abstract is meant to be a summary of the article. It is NOT part of the paper (...but summarizes it). Within the abstract, please give a few more details on the classification and the relevant properties. (Re-)move details on the methods, if they are not relevant within the summary. For example, the information on the 1-50Hz frequency band is currently only given in the abstract, and not in the paper.

We propose a new abstract taking into account the comments of the editor. We remove the 1-50Hz information from the abstract.

Discussion: the discussion currently contains many awkward phrases and could

be clearer. I encourage you to carefully look through this section and try to improve it.

We have rewritten partly the discussion.

1.11 space missing between 'proposed:' and 'slopequake'

Corrected.

1.11 I find the use of the word 'gathering' in this sentence unclear. Can you be more clear?

We replace by " correspond to".

1.12 Please give a brief summary of the difference between Rockfalls and Granular Flows in the abstract.

We added this information in the abstract.

1.16 The last sentence should not be part of the abstract and should be deleted.

We have deleted accordingly.

4.21 . . . variety of signals is observed. . .

Corrected.

6.4 it might be good to repeat the definition of AE at this point

We have changed accordingly.

7.15 maybe change to '. . . can be very challenging. . . '

We have changed accordingly.

10.21 Somewhere in this chapter, include the information of the 1-50Hz frequency band.

We kept the correct information : [$f_c$-50] Hz band.

11.6 '...average of on a moving window. . . ' – delete 'of'

Corrected.

12.4-10 how does this fit with the information that you concentrate on the 1-50Hz frequency band, as mentioned in the abstract? Please clarify.

We answered previously. Please, see l10.21 and first response.

13.1 break

13.5 . . . and in the data reported by Hibert et al. (2011) and Dietze et al. (2017b).

13.7 . . . scaling laws have also been established. . .

13.21 Space missing before 'The dissymmetry. . . '

13.25 'increase'

13.26 'rapidly'

15.10 '. . . it remains. . . '

17.2 '. . . can be differentiated. . . '

17.4 '. . . more examples. . . '

17.11 'Colombero et al. (2018) suggested. . . '

17.16 '. . . results of Colombero et al. (2018).'

17.20 '. . . rock fall signals. . . '

All the above mistakes have been corrected.

17.21 what does the number 12 refer to in the reference to Super-Sauze?

The number refers to a figure, we added the missing "Fig.".

17.22 '. . . characterized. . . '

17.28 '. . . and Super-Sauze landslides (Vouillamoz et al., 2017).'

17.34 'Gomberg et al. (2011) hypothesized. . . '

18.5 'Systematic location of these events is needed. . . must be integrated. . . in the case that they. . . '

18.15 'Similar tremors have been found. . . '

All the above mistakes have been corrected.

18.28 unclear – what is meant by instabilities deformation?

We mean "instabilities deformation field at the surface".

18.29 '. . . currently prevent. . . '

18.30 (see also 18.23) either 'First. . . second. . . ' or 'Firstly. . . secondly. . . '

18.33 '. . . which remain. . . '

19.25 '. . . three main classes. . . '

20.8 clean up citations

Figure 5: typo in one of the location names; the caption misses a dot after Fig. (this repeats in other captions)

All the above mistakes have been corrected.

[revised manuscript text omitted]